# ASSESSING LARGE LANGUAGE MODELS ON CLIMATE INFORMATION

## ABSTRACT

Understanding how climate change affects us and learning about available solutions are key steps toward empowering individuals and communities to mitigate and adapt to it. As Large Language Models (LLMs) rise in popularity, it is necessary to assess their capability in this domain. In this study, we present a comprehensive evaluation framework, grounded in science communication principles, to analyze LLM responses to climate change topics. Our framework emphasizes both the presentational and epistemological adequacy of answers, offering a fine-grained analysis of LLM generations. Spanning 8 dimensions and 30 distinct issues, the task is a real-world example of a growing number of challenging problems where AI can complement and lift human performance. We introduce a novel, practical protocol for scalable oversight that relies on AI Assistance and raters with relevant education. We evaluate several recent LLMs on a set of popular climate questions. Our results point to a significant gap between surface and epistemological quality of LLMs in the realm of climate communication.

## 1 INTRODUCTION

As concerns around *climate change* continue to intensify (Poushter et al., 2022; WHO, 2021), more and more people turn to digital media as their primary source of information (Newman et al., 2021). However, in spite of ubiquitous access to information, there remains a considerable gap in public climate literacy, exacerbated by the spread of mis- and disinformation (Leiserowitz et al., 2022). The challenge of conveying climate data arises from the nature of scientific communication itself: science, as an evolving domain, is laden with specialized knowledge, technical complexity, and inherent uncertainties (Moser, 2016). The digital media landscape, characterized by limited attention spans and adversarial dynamics, further compounds these challenges (Pearce et al., 2019).

We explore the potential of AI in curating and presenting climate information in an accessible manner. While AI's promise in addressing global climate challenges is evident through its applications in climate modeling, energy optimization, and disaster management (Rolnick et al., 2022), its intersection with Natural Language Processing (NLP) is still under-explored. Recent advancements in LLMs (Brown et al., 2020; Chowdhery et al., 2022; OpenAI, 2023) have captured the attention of the scientific community and the general public for their performance on standard benchmarks, and their broad approachability as information technology. Given their tremendous potential, there is hope that LLMs may support us in addressing climate information challenges.

However, using LLMs to address science-related information needs raises safety concerns, due to their limitations in assessing factuality (Weidinger et al., 2021; Birhane et al., 2023). Fluent, grammatical responses and advanced linguistic dialogue behaviors are preferred and trusted by users, even in the absence of trustworthy information (Chiesurin et al., 2023). This makes evaluating LLMs, especially with non-expert human raters, treacherous. Research on how to evaluate systems that may achieve or exceed human abilities, or *scalable oversight* (Amodei et al., 2016) is so far mostly theoretical (Irving et al., 2018; Leike et al., 2018; Christiano et al., 2018).

Our work contributes to this growing field. We have meticulously developed a framework based on Science Communication research, tailored to the responses of LLMs within the climate change context. Research points out the importance of how information is presented (Jamieson et al., 2017). Drawing on the wealth of scientific knowledge, we examine relevant principles and best practices to propose an implementation of a human assessment framework that delivers high-quality results with

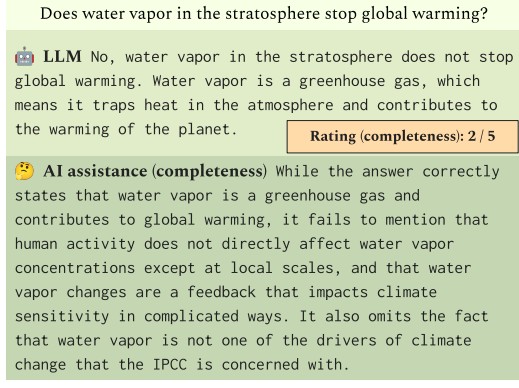

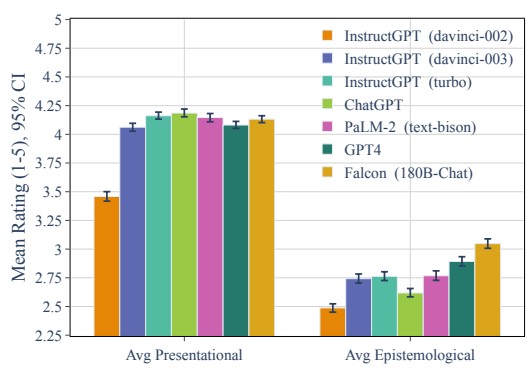

(a) Example of a question, answer, AI Assistance and human rating.

(b) Aggregated (simple average) presentational and epistemological ratings.

Figure 1: Rated example and average results for several LLMs.

educated (but non-expert) raters. We present results from an empirical study on a set of 300 questions concerning popular, as well as long-debated, topics on climate change. We systematically assess **presentational** properties such as *style*, *clarity*, linguistic *correctness*, and *tone*. More importantly, we also assess **epistemological** issues: *accuracy*, *specificity*, *completeness*, and *uncertainty*.

Our main contributions are as follows: (1) We introduce a principled evaluation framework for LLMs on climate information,[1] developed through a rigorous interdisciplinary approach. (2) To improve rating performance, we introduce a novel protocol for scalable oversight that uses AI Assistance (cf. Figure 1a) and relies on raters with relevant educational background. (3) Our experiments involve the most recent and prominent LLMs to demonstrate the relevance of the evaluation. (4) Results (Figure 1b) suggest that, while exceptionally fluent, current LLMs have much room for improvement regarding content quality on climate information. Thus, our framework provides concrete directions for improving future LLMs for communicating scientific information. (5) Finally, we analyze the relation of these dimensions to attribution-based evaluations of LLMs (Rashkin et al., 2022) and find that they emerge as mostly orthogonal and complementary aspects.

We also emphasize that our experiments are the first on this topic. As such, our analysis and its conclusions are limited to the current setting. For instance, we don't apply model-specific prompt engineering or reasoning methods, which are known to be effective. More generally, we highlight two challenging questions that this work leaves unanswered: 1) the way individual dimensions should be combined in single metrics, e.g., for model selection, and, 2) how to model and alleviate the bias introduced by the AI Assistance. The former problem involves value-based choices that may require a task-specific setting. The latter will be one of the main focuses of our future work.

## 2 EVALUATIVE DIMENSIONS FOR CLIMATE INFORMATION

Scholarship on science communication – originating from disciplines such as communication science, sociology, psychology, human geography, and education, among others (Trench & Bucchi, 2021; Nisbet et al., 2018; Jamieson et al., 2017) – offers conceptual arguments and empirical evidence for appropriately disseminating scientific information, e.g., on climate change, to the general public (König et al., 2023; Lewis Jr. & Wai, 2021). Two basic dimensions have to be distinguished here. (1) Presentational features of the messages that contain the information, such as their comprehensibility (Lang, 2000), to ensure that recipients can receive, understand, memorize, and retrieve such information. We conceive this dimension as *presentational adequacy*. (2) The conveyed information must represent the current state of scientific knowledge as adequately and comprehensively as possible while being specific and appropriately communicating associated uncertainties (Fähnrich et al., 2023). We conceive this dimension as *epistemological adequacy*.

---

[1]To aid reproducibility, we provide the exact evaluation protocols and all prompts used to generate data.

## 2.1 PRESENTATIONAL ADEQUACY

An adequate *presentation* should comply with three criteria (Jamieson et al., 2017): (1) be comprehensible, (2) aid understanding through layout and visualizations, and (3) use appropriate sources and references. Here we focus primarily on comprehensibility. We return to sources and references in Section 4 and discuss visualization in Section 5. The *comprehensibility* of a text can be conceptualized along four criteria: style, clarity, linguistic correctness, and tone.

**Style.** The language style should not be too informal or colloquial (Mazer & Hunt, 2008), as this can undermine the credibility of information and cause users to rely on their own rather than expert judgements (Scharrer et al., 2012). Moreover, texts should not be too short, because exposure to brief snippets of scientific information may lead recipients to get a "feeling of knowing" from reading messages that contain insufficient information (Leonhard et al., 2020). Long texts, however, require high motivation and cognitive resources that readers may not want to invest (Lang, 2000). In addition, some stylistic dimensions can be borrowed from the Multidimensional Quality Metrics (MQM) framework, which was designed to assess the quality of (machine) translated texts (Lommel et al., 2013). One of the MQM's core dimensions is 'terminology', referring to the correct and consistent use of (in this case scientific) terminology.

**Clarity.** Climate-related messages should be clearly formulated (Maibach et al., 2023). Risk and health communication research also support the efficacy of concise and clear language, as less detailed texts require less cognitive effort and are preferred by users (Fagerlin & Peters, 2011; Neuhauser & Paul, 2011). The use of jargon should be avoided (Baram-Tsabari & Lewenstein, 2013; Baram-Tsabari et al., 2020), as technical terms can inhibit readers' ability to process information (Bullock et al., 2019; Brooks, 2017; Shulman et al., 2020). Clarity seems particularly relevant for individuals with lower numeracy skills (Bruine de Bruin & Bostrom, 2013). If numbers are used, the presentation should be tailored to the recipient's numeracy level (Fagerlin & Peters, 2011).

**Correctness.** MQM (Lommel et al., 2013) emphasizes that messages should adhere to linguistic quality criteria to be comprehensible: One of its core components is adherence to linguistic conventions, i.e., the correct use of punctuation, spelling, and grammar.[2] Violating these criteria can damage the perceived credibility of the message or its sender (Berger, 2020) and has been shown to influence behavior (e.g., Mollick, 2014). Accordingly, linguistic correctness is an important aspect of the presentational adequacy of science communication (Mercer-Mapstone & Kuchel, 2017).

**Tone.** The tone of a message is essential. It concerns the neutrality of the tone, its persuasiveness and its positivity or negativity. Messages should not adopt or lean towards a certain valence, worldview, or ideological conviction in order to be effective (Blanton & Ikizer, 2019; Yuan & Lu, 2020). In climate-related messages a neutral tone can be more effective than a persuasive tone (Kerr et al., 2022; Munoz-Carrier et al., 2020). Likewise, messages should not use too positively or negatively valenced language, particularly if the goal is to convey factual information (Palm et al., 2020).

## 2.2 EPISTEMOLOGICAL ADEQUACY

The epistemological adequacy of climate-related messages is of greatest importance. This entails several aspects: (1) accuracy, (2) specificity, (3) completeness, (4) the degree of (un)certainty, and (5) the presentation of methods and methodology. We focus on the first four dimensions here, leaving the latter for future work (cf. also the discussion in Section 5).

**Accuracy.** A basic principle is that scientific information – such as climate change information presented by LLMs – should be *accurate* (Kelesidou & Chabrol, 2021). *Incorrect, wrong,* or *self-contradictory* information that takes scientific findings or anecdotal evidence out of context should be prevented. (Hinnant et al., 2016). This is particularly important when considering known accuracy issues of LLMs such as *hallucination* (Schäfer, 2023; Ji et al., 2023).

**Specificity.** Information that is important to the audience should not be missed, while ignoring irrelevant information. Communication should address the regional and temporal contexts of target audiences. In other words, it should be *relevant* to the respective audience, i.e., should fit their personal contexts *spatially and temporally*. Research shows that specific, local information leads to a higher perceived relevance among recipients (Leiserowitz & Smith, 2017; Lee et al., 2015). For

---

[2]https://themqm.info/typology

an answer to have high temporal fit, it should address the time frame mentioned in the question. For questions where a specific time frame is not specified, the answer should generally be based on information and data that is up to date. Research has also shown that "here & now" associations can be powerful in science communication (Holmes et al., 2020).

**Completeness.** Answers should be *complete*. Rather than only referring to a part of the question posed, the answer should be formulated in a way that addresses all parts of the question in full (Bergquist et al., 2022; Leiserowitz & Smith, 2017). At the same time, to answer all aspects of the question, the information given should reflect the depth and breadth of relevant scientific knowledge available regarding the topic(s) addressed by the question (Kelesidou & Chabrol, 2021).

**Uncertainty.** Communicating the level of agreement and certainty for scientific findings can be crucial to adequately informing the audience (Budescu et al., 2012; Howe et al., 2019). Likewise, when the level of agreement or quantified certainty is unknown, the audience should be informed about the uncertainty and/or isolation of the supporting evidence (Keohane et al., 2014). This is particularly important in climate communication (Chinn & Hart, 2021; Goldberg et al., 2022; Maertens et al., 2020), as the scientific consensus on climate change has been found to function as a "gateway belief", implying that perceived scientific agreement can positively influence the belief in human-caused climate change and motivate public action (van der Linden et al., 2015).

## 3 PRESENTATIONAL AND EPISTEMOLOGICAL ADEQUACY EVALUATION

### 3.1 SCOPE

We experiment with our evaluative dimensions using a human rating framework. A comprehensive evaluation of LLMs on climate information should cover a broad spectrum of information needs, including the basics of climate science, mitigation, adaptation etc. Context-specific information needs are also necessary; e.g., to address the concerns of more vulnerable or under-resourced communities. This is not an easy problem as no standardized tests exist to assess climate-related knowledge. We begin by focusing on a set of popular topics and long-debated questions. For this purpose, we turn to search engines, popular climate forums and Wikipedia. These are by no means the most important, or most useful, or hardest questions on the topic. But they provide a reasonable starting point that allows us to build a framework that can hopefully support delving into deeper topics.

### 3.2 DATA

**Questions.** We collect a diverse set of 300 questions from three different sources. For the first set, we use `GPT-4` to generate questions from the English Wikipedia articles. First, we select articles that are related to climate change, then we feed in the paragraphs of each of the selected articles to `GPT-4` and task the model to generate questions that can be answered by the paragraph. For the second set, we turn to Skeptical Science, a website that publishes authoritative information about climate science. We take the list of debated *myths*[3] and manually rephrase them as questions. For the third set, we use Google Trends, a tool that provides data on public interest in specific search terms and topics.[4] We collect the most popular questions, by search volume, from the U.S., for the topics 'Climate Change' and 'Global Warming'. We post-process all questions to remove duplicates, questions that are not related to climate change, and questions that are taken out of context. Finally, we sample 100 questions from each set. Please see Appendix C.1 for the details.

**Answers.** There are many ways to get answers from LLMs. The quality can vary greatly depending on prompt engineering, reasoning schemes, in-context learning etc. However, a straight question is the most common way for users to get answers from LLMs. As such it provides a solid baseline, reducing variance due to individual LLM's skills and the challenge of finding local optima in the absence of pre-existing data. Thus, we intentionally focus on a plain prompt, leaving more advanced techniques to future work. We prompt each LLM with the instruction: *You are an expert on climate change communication. Answer each question in a 3-4 sentence paragraph.* We include the answer length information to anchor the expected answer form on an objective value, thus reducing answer structure variance and possible cascading confounding factors at evaluation time.

---

[3] https://skepticalscience.com/argument.php
[4] https://trends.google.com/trends/.

**Keypoints.** To find supporting evidence for an answer, for AI Assistance and attribution evaluations (Section 4), we extract keypoints from each answer. To do so, we instruct `GPT-4` to examine all the statements in the answer, and identify 1 to 3 key statements that are made to answer the question.

**Evidence.** We fetch evidence for each keypoint in the answer. Given the question and the answer, we first ask `GPT-4` to provide URL(s) of Wikipedia articles that support the answer. We limit evidence to Wikipedia because `GPT-4` is fairly consistent in generating relevant, valid Wikipedia URLs, while the quality is lower for other web sources. Furthermore, Wikipedia is uniform in style and quality as it adheres to established guidelines. We break down each article into its paragraphs. For each keypoint, we ask the model to rank the paragraphs based on their relevance to the keypoint and the question, and pick the 3 highest ranking as the evidence. Table 7 shows an example. We found that using keypoints, in combination with URL generation and evidence selection, is a simple and effective solution, compared to a paired retrieval components, for our current setup.

**AI Assistance.** To assist human raters, we use `GPT-4` to generate assistance along the dimensions introduced in Section 2. For each dimension, we ask the model to express its agreement or disagreement that the information is presented well according to that dimension. For epistemological dimension, we also provide the retrieved evidence and instruct the model to use that verbatim to support its disagreement (if any). Please refer to Table 4 for a complete list of prompts used to generate the data, and to Appendix E for some statistics of the generated answers.

## 3.3 RATING FRAMEWORK AND RATERS

Our rating task involves evaluating an answer based on the four presentational (Section 2.1) and the four epistemological dimensions (Section 2.2). Screenshots of the template can be found in Appendix M.4. We select candidate raters with relevant educational background (see Appendix M.1). To be admitted to the task, after finishing a brief tutorial, the raters need to pass an admission test that evaluates their performance on three full examples (see Appendix M.3). A summary of the broad demographics of raters that participated can be found in Appendix M.1. Each answer is assessed by three human raters. We compute agreement metrics for all experiments and report the numbers in Appendix H. We don't forbid or discourage consulting external sources, but specify that this should be done nimbly, to clarify specific points and not for extensive research.

## 3.4 EXPERIMENTAL RESULTS

In Figure 1b we provide a synthetic view of the results by averaging within the presentational and epistemological dimensions, to illustrate the main take-home that LLMs overall are much better in the presentational dimensions. Otherwise, we only report individual dimensions result. We intentionally avoid suggesting ways of combining the individual dimensions into single metrics. How to combine the scores in appropriate ways is an important, and non-trivial, problem for making decisions, but it's besides the scope of this work.

**High-level view.** Figure 1b provides an overview of the rating results for the following LLMs: `GPT-4` (OpenAI, 2023), `ChatGPT-3.5`, `InstructGPT` (turbo), `InstructGPT` (text-davinci-003), `InstructGPT` (text-davinci-002)[5], as well as `PaLM2` (text-bison) (Anil et al., 2023) and `Falcon-180B-Chat`[6]. For a full summary of results, for all the individual dimensions, see Figure 2.

All models, except for `InstructGPT` (text-davinci-002), perform well on presentation (Figure 1b and Table 1). This demonstrates how far LLMs have come in terms of surface form quality, seemingly after the introduction of learning from human preferences (Ouyang et al., 2022). We note, however, a marked performance drop for *tone* (cf. Figure 2). This dimension captures more subtle challenges for LLMs, touching on aspects related to Pragmatics (Levinson, 1983). Table 23 shows an example, while Appendix B elaborates on the subject in the broader context of argumentative style.

The epistemological evaluation reveals lower performance on all systems. Except for *accuracy* (Figure 2), performance is consistently below average, especially for *completeness* and *uncertainty*. We also note that the latter epistemological dimensions may be difficult to satisfy in short 3-4 sentence answers. Being comprehensive in such a short space is harder than being accurate. On the other

---

[5]`https://platform.openai.com/docs/models`.
[6]`https://falconllm.tii.ae/falcon.html`.

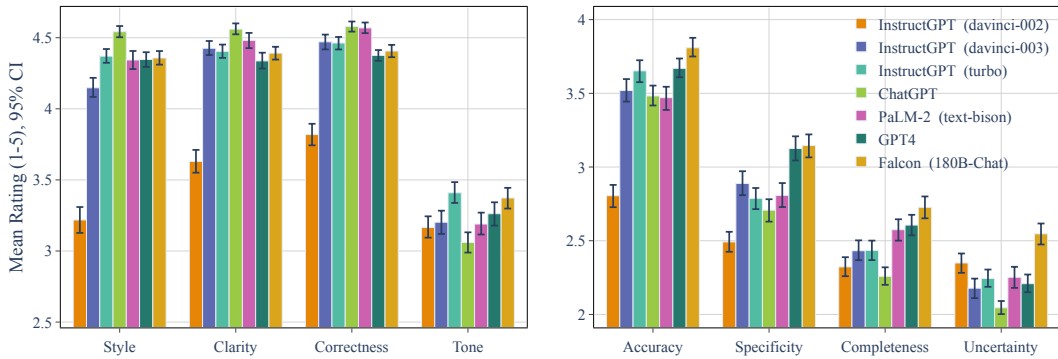

Figure 2: Results for all presentational and epistemological dimensions.

hand, we notice that LLMs can also waste space on generic statements (cf. Appendix B). Overall, on climate information, current top-of-the-line LLMs have significant headroom for improvement. For examples, please see Tables 24 to 27. Table 2 reports complete results and confidence intervals.

**Resolution and Range.** Our evaluation has often sufficient resolution to tell models apart on specific dimensions, indicate where they differ and suggest trends. For instance, ChatGPT is the best amongst the LLMs tested in the presentation dimensions, except *tone*, but ranks between position 4 and last on the epistemological scores (Figure 2). This brings up the hypothesis of tradeoffs between presentational and epistemological properties. In fact, GPT-4 is always better than ChatGPT on the epistomological evals but worse on most presentational ones. Noticeably, the most competitive model on the epistemological dimensions is a recent open model, Falcon-180B-Chat. Falcon-180B-Chat's performance may be related to its large size, but we can only speculate as this information is not generally available for all models. More generally, the difference between the best LLM and the worst in specific dimensions – e.g., Falcon-180B-Chat and InstructGPT (text-davinci-002) on the epistemological ones, and, respectively, ChatGPT and InstructGPT (text-davinci-002) on the first three presentational ones – is large compared to the estimated standard deviation, providing evidence that the evaluation has sufficient dynamic range.

**Impact of AI Assistance.** Raters should identify more (real) issues with assistance, because it may make them aware of additional issues. We find supporting evidence in two separate experiments.

Figure 3a reports the number of issues detected for each dimension on GPT-4 answers in three different settings, each with a different degree of the raters' exposure to assistance. The setting 'Without AI Assistance' refers to a setting where a pool of raters is never provided with assistance. The second setting 'Without AI Assistance, but previous exposure' refers to a setting where no assistance was shown, but the raters have worked on previous studies that included assistance.[7] Lastly, 'With AI Assistance' denotes the standard setting where assistance is shown anytime is available. Results suggest that the presence of assistance is key for detecting more issues. This is consistent with the results from Saunders et al. (2022), who found improved rater performance with assistance. Raters with extensive previous exposure to assistance are in an interesting "middle" position: They detect more issues than the assistance-unaware group, but less than the group provided with specific assistance for the experiment. This suggests that raters learn from repeated exposure to assistance, and show improved performance even when no assistance is present.

Further evidence of the usefulness of AI Assistance comes from our validation experiments (cf. Appendix G for more details). Similar to Saunders et al. (2022), we want to determine if assistance helps surface real issues, without general access to gold truth in our data. To do this, the authors manually generated 30 different examples, each exhibiting a particular issue. We found that the majority of three raters detected 77% of issues when shown assistance, while the majority of three raters only detected 60% of the issues when not shown assistance. The data we collected on the

---

[7]We do make sure that the raters have not worked on the same examples before and have never seen assistance for the specific examples they are working on.

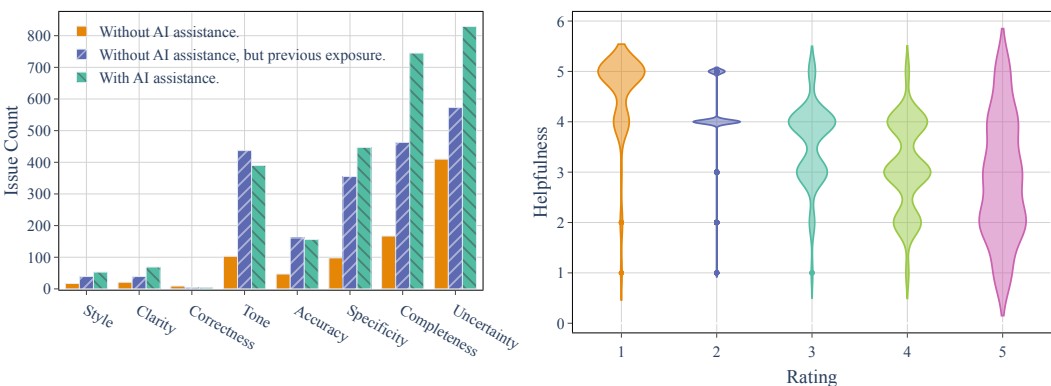

(a) Number of issues detected depending on AI Assistance exposure.

(b) The relationship between rating and reported helpfulness of the AI assistance (on the same scale).

Figure 3: Evidence of the impact of AI Assistance.

helpfulness of assistance suggests that when raters do not find assistance helpful, they give higher ratings (see Figure 3b). This indicates that the raters can think critically about the assistance and do not follow it blindly. These experiments provide evidence that the AI Assistance helps the raters find real issues that they would not have otherwise been discovered.

**Other Findings.** Comparing the rating outcome by source of the question – Skeptical Science, GTrends, and synthetic questions based on Wikipedia paragraphs – we find no major differences, with a slight trend that Wikipedia questions tend to be more specific and thus harder to answer. In particular, we see no evidence that GPT-4 performs better on questions that were generated with GPT-4 compared to the other sources. Similarly, the topic of the question does not show a strong correlation with answer quality. See Appendix I for additional discussion and figures. In preliminary experiments, we also find that describing the evaluation criteria in the prompt can improve performance on the difficult dimensions, cf. Appendix A.1. Interestingly, this comes at the cost of degraded performance on the presentational dimensions.

## 4  EPISTEMOLOGICAL ADEQUACY AND ATTRIBUTION

Audiences of science and climate communication are more likely to trust information if the source is perceived as credible, engaged and concerned about the audience's interests (Brown & Bruhn, 2011; Maibach et al., 2023; Hayhoe, 2018). An adequate presentation of climate information should include curated references. To address the factuality limitations of LLMs, researchers have proposed Attribution to Identified Source (AIS) as a dedicated evaluation (Rashkin et al., 2022; Dziri et al., 2022). An attributable answer must include an explicit quote, from an existing document, in order to support its claims and reduce hallucination (Menick et al., 2022; Bohnet et al., 2023).

Evaluating the ability of LLMs to properly reference the statements they make goes beyond the scope of this paper. For instance, as proposed by Liu et al. (2023), this may involve evaluating generative search engines. However, we started examining the relationship between attribution and the epistemological dimensions with an AIS experiment. We run this experiment only on GPT-4. In our data, each answer is associated with a set of keypoints which, in turn, are used to identify Wikipedia articles that are likely to contain supporting evidence. For 87.7% of the questions, GPT-4 produces a valid Wikipedia article from which evidence passages can be extracted. We evaluate the attribution of each keypoint individually by asking the annotators whether a keypoint is fully, partially or not supported by the evidence. 66.79% of keypoints are either fully or partially supported. At the answer level, 46.08% of the answers are fully or partially supported by the evidence. While far from perfect, the data suffices for a first analysis (cf. Appendix F for details).

Figure 4 compares the distribution of average epistemological ratings, with respect to the attribution of answers, revealing interesting trends. In both the *accuracy* and *specificity* dimensions, we observe that answers that are fully attributed have higher minimum ratings compared to answers

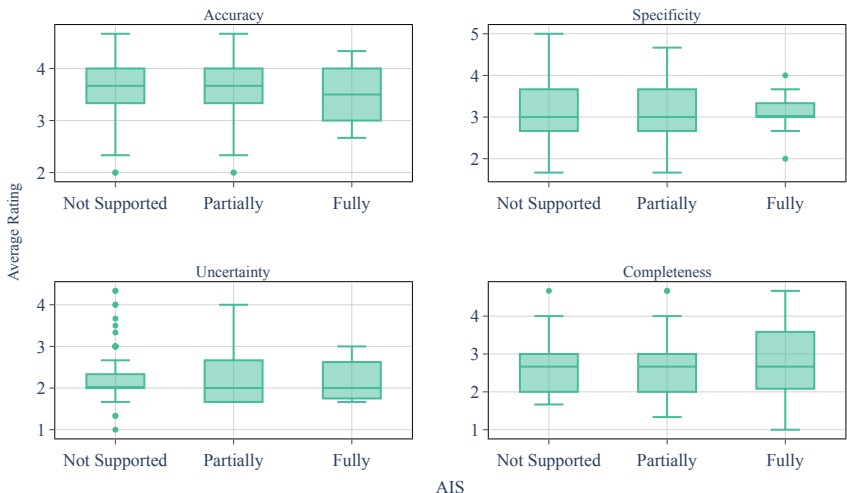

Figure 4: Comparing AIS ratings with average ratings of the 4 epistemological dimensions.

that are only partially attributed, or not attributed at all. Interestingly, we see an opposite pattern in the *completeness* dimension: Answers that are fully attributed have lower minimum ratings on *completeness*. This result highlights a blind spot for attribution methods; AIS can only consider what *is* included in the answers, and not what important information is missing. In the *uncertainty* dimension, we observe that there are more answers with low uncertainty ratings among the answers that are not attributed, compared to answers that are either partially or fully attributed.

More generally, there does not seem to be any correlation between AIS and epistemological results. The Spearman's coefficient between AIS and the 3-raters mean rating value for *accuracy*, *specificity*, *uncertainty* and *completeness* are, respectively: $0.03, -0.06, 0.002, -0.02$, with corresponding p-values: $0.65, 0.31, 0.97, 0.78$. We interpret this as evidence that AIS and epistemological assessments are orthogonal and complementary. We provide more qualitative support in Table 8. At a high level, this suggests that attribution, either human or model-based, is not a reliable proxy for epistemological quality. On the other hand, grounding in authoritative sources is required of good science communication. We leave it to future work to extend our framework to include references in a principled way.

## 5 LIMITATIONS AND FUTURE WORK

Our rating dimensions inherently have a subjective component, introducing noise when evaluating at the answer-level. However, our findings suggest that the evaluation is robust at the system level. Another limitation of our work is that we do not have access to gold ratings. As procuring reliable human judgements becomes unfeasible and/or uneconomical, especially for complex and difficult tasks, such a setting is likely to become more common in the future. Hence, this poses an exciting challenge for future studies, and we envision evaluation frameworks of the kind proposed here serving as a valuable testbed to develop new protocols for *scalable oversight*. A related topic is the role of LLMs as raters. Preliminary experiments are promising (Appendix L). We found that, as with humans, LLMs benefit from AI Assistance and that humans and LLM raters tend to agree on major points. What bias gets introduced by assistance (and rating), and how to measure and control it properly, is a significant open question that needs to be addressed. This links this research to the broader AI alignment field. How to model and alleviate the bias introduced by the AI Assistance will be one of the main focuses of our future work.

Ideally, an answer would be tailored towards the audience, and take into account their specific attributes (Hendriks et al., 2016; Klinger & Metag, 2021). Unless specifically prompted, LLMs do not do this. We explore in Appendix B how the kind of arguments LLMs seem to gravitate towards may hurt their efficacy with some audiences, and leave further exploration to future work. Research also provides abundant evidence on the importance of supplementing textual information with visual

aids in the form of cartoons, charts, pictographs and videos (Flemming et al., 2018; Brown & Bruhn, 2011). Visual complements can be especially useful for understanding quantitative data (Fagerlin & Peters, 2011) and in the case of limited literacy (Wolf et al., 2010). The abstract nature of climate change, and its distant implications, makes visualization particularly challenging (Schäfer, 2020).

## 6    RELATED WORK

**Evaluating LLMs.** While LLMs can generate fluent text, responses are not always adequately grounded, attributable to reliable sources, and complete. For instance, Liu et al. (2023) assess four generative search engines and report that, although responses are perceived as high quality, only half are fully supported. Their findings reveal an inverse correlation between fluency/utility and evidential support. Xu et al. (2023) advocate for expert-level human evaluations in question answering, cautioning against over-reliance on single metrics instead of comprehensive assessments. Another domain that needs expert-level evaluation is the medical domain. Singhal et al. (2023) propose Med-PaLM, an LLM for medical information, and introduces a clinical evaluation framework.These cover criteria like alignment with scientific consensus, potential harm, and comprehension. Evaluating LLMs on climate information is another domain that can benefit from expert-level evaluation. However, prior work mainly emphasizes text classification (Diggelmann et al., 2020; Varini et al., 2020) and sustainability report analysis (Webersinke et al., 2022; Bingler et al., 2022). This study aims to fill this gap by providing a comprehensive evaluation framework for climate change.

**Scalable Oversight.** This area, introduced by Amodei et al. (2016), studies the question of how to scale human oversight, especially in the setting where evaluating (or supervising) models becomes increasingly difficult. Contributions have initially focused on theoretical proposals for how AI can help humans supervise models that exceed their abilities (Irving et al., 2018; Leike et al., 2018; Christiano et al., 2018). Following Irving et al. (2018), one can see our AI Assistance as a single-turn debate, where the human annotator is shown the answer proposed by the model and a single response to that answer.[8] Two recent studies provide interesting proofs of concepts for AI Assistance: Bowman et al. (2022) study *sandwiching*, an approach where non-experts align a model with the help of a model while experts provide validation. They show that non-expert raters perform better on an (artificially) difficult multiple-choice task when interacting with a dialogue agent. Saunders et al. (2022) report that human raters of summarization tasks produce more critiques when given the opportunity to accept or edit critiques written by a model. Our work contributes a study of a *scalable oversight* protocol to improve rating quality in a realistic setting.

**AI Ratings.** Recent studies explore the feasibility of evaluations performed by AI. Kocmi & Federmann (2023) indicate that LLMs can perform state-of-the-art quality assessment of translations, even without references. Their work has been extended to automatic MQM annotation by Fernandes et al. (2023). Gilardi et al. (2023) reports that ChatGPT has a higher agreement with expert-level raters than with less qualified ones. Chiang & Lee (2023) argue that humans and LLMs ratings are correlated but point out LLM's factuality and bias limitations. Instead of replacing human raters entirely, in our work we demonstrate the effectiveness of using AI Assistance to aid educated raters.

## 7    CONCLUSION

We introduce an evaluation framework informed by science communication research and assess LLMs on a first set of common climate information needs. The task is difficult for human raters. To support them, an important part of our framework relies on a novel and practical protocol for scalable oversight that leverages AI Assistance. It is important to realize that these are the first results of this kind and more research is needed. In particular: we cover only a small set of general information needs, and while there is evidence that AI Assistance is valuable in evaluating LLMs on such tasks, we need to develop a framework to understand and mitigate the bias it carries. Overall, our results suggest that, while presentationally adequate, current LLMs have much room for improvement regarding the epistemological qualities of their outputs. More research is needed to improve these aspects of LLMs. We hope our framework will not only directly be used to evaluate LLMs on climate information, but also inspire other researchers to come up with better and more principled evaluations in general.

---

[8]In the setting of Irving et al. (2018), this corresponds to the second level of the polynomial hierarchy $\Sigma_2^P$.

ETHICS STATEMENT

The details of our study design, including compensation rates, were reviewed by an independent ethical review committee. All raters provided informed consent prior to completing tasks and received fair compensation with respect to local markets. It is our policy that researchers must pay workers/participants at least the living wage for their location. No personally identifiable information (PII) was collected or will be released.

We conducted the experiments in English, therefore we do not claim generalization of our findings across languages. However, we believe that the proposed methods could be transferred to other languages.

LLMs are already an important source of information for many people, and it is important to assess whether they can adequately address information needs around climate change. Our work contributes to this effort and sheds light on both the potential and the limitation of LLMs in this domain.

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

APPENDIX

# A  MAIN RESULTS

See Figure 1b for a comparison along the main dimensions and Tables 1 and 2 and Figure 2 for detailed results for all evaluated models.

| System | style | clarity | correctness | tone |
|---|---|---|---|---|
| ChatGPT | 4.54 [4.50, 4.58] | 4.56 [4.52, 4.60] | 4.58 [4.54, 4.61] | 3.06 [2.99, 3.13] |
| InstructGPT (davinci-003) | 4.15 [4.08, 4.22] | 4.43 [4.38, 4.47] | 4.47 [4.42, 4.52] | 3.20 [3.12, 3.28] |
| InstructGPT (davinci-002) | 3.22 [3.13, 3.31] | 3.63 [3.55, 3.70] | 3.82 [3.74, 3.90] | 3.17 [3.09, 3.24] |
| InstructGPT (turbo) | 4.37 [4.32, 4.42] | 4.40 [4.36, 4.45] | 4.46 [4.42, 4.51] | 3.41 [3.33, 3.48] |
| PaLM-2 (text-bison) | 4.34 [4.28, 4.40] | 4.48 [4.43, 4.53] | 4.57 [4.53, 4.61] | 3.19 [3.11, 3.27] |
| GPT4 | 4.35 [4.30, 4.40] | 4.34 [4.28, 4.39] | 4.38 [4.34, 4.41] | 3.26 [3.19, 3.34] |
| Falcon (180B-Chat) | 4.36 [4.31, 4.41] | 4.39 [4.35, 4.44] | 4.41 [4.36, 4.45] | 3.37 [3.30, 3.45] |
| GPT4, no assistance, prev. exposure | 4.59 [4.54, 4.63] | 4.63 [4.59, 4.68] | 4.66 [4.63, 4.70] | 3.24 [3.16, 3.32] |
| GPT4, no assistance | 4.45 [4.41, 4.50] | 4.57 [4.53, 4.61] | 4.74 [4.70, 4.77] | 4.35 [4.29, 4.42] |

Table 1: Results along the presentational dimensions, with $95\%$ confidence intervals.

| System | accuracy | specificity | completeness | uncertainty |
|---|---|---|---|---|
| ChatGPT | 3.48 [3.41, 3.55] | 2.71 [2.63, 2.78] | 2.26 [2.20, 2.31] | 2.05 [2.00, 2.09] |
| InstructGPT (davinci-003) | 3.52 [3.44, 3.60] | 2.89 [2.81, 2.97] | 2.43 [2.36, 2.50] | 2.18 [2.11, 2.25] |
| InstructGPT (davinci-002) | 2.81 [2.73, 2.88] | 2.49 [2.42, 2.56] | 2.32 [2.26, 2.39] | 2.35 [2.29, 2.41] |
| InstructGPT (turbo) | 3.65 [3.58, 3.73] | 2.79 [2.71, 2.86] | 2.43 [2.37, 2.50] | 2.24 [2.19, 2.30] |
| PaLM-2 (text-bison) | 3.47 [3.39, 3.55] | 2.81 [2.73, 2.89] | 2.57 [2.50, 2.65] | 2.25 [2.18, 2.32] |
| GPT4 | 3.67 [3.61, 3.73] | 3.13 [3.05, 3.21] | 2.61 [2.53, 2.68] | 2.21 [2.15, 2.27] |
| Falcon (180B-Chat) | 3.81 [3.74, 3.87] | 3.15 [3.07, 3.23] | 2.73 [2.65, 2.80] | 2.55 [2.47, 2.62] |
| GPT4, no assistance, prev. exposure | 3.86 [3.79, 3.93] | 3.43 [3.35, 3.52] | 3.30 [3.21, 3.39] | 2.78 [2.69, 2.87] |
| GPT4, no assistance | 4.49 [4.44, 4.55] | 4.41 [4.35, 4.48] | 4.32 [4.25, 4.39] | 3.38 [3.29, 3.46] |

Table 2: Results along the epistemological dimensions, with $95\%$ confidence intervals.

## A.1  DIMENSION-AWARE PROMPTS

In our base case scenario we use the simplest question answering prompt. One interesting question concerns the inclusion of our evaluation criteria in the instructions; e.g., provided as part of the prompt. We run a preliminary experiment in this mode by adding a description of the criteria to the prompt for GPT-4:

> You are an expert on climate change communication. Answer the question in a 3-4 sentence paragraph. The answer should be concise and tailored for a general audience. It must be clear, and easy to understand. The answer should be presented in a neutral, unbiased tone without any negative connotations or attempts to persuade. The answer should be factually accurate. The answer should be specific to the question and avoid irrelevant, generic, or vague statements. The answer should comprehensively address all aspects of the question. Where scientific uncertainty exists, the answer should appropriately reflect this, conveying the range of scientific perspectives or the limitations of current knowledge.

As a control we re-run the base prompt experiment with GPT-4. The results are summarized in Table 3.

First, we notice that, even with the base prompt, the performance of GPT-4 has improved across the board with respect to the results in Table 1 and Table 2. This is probably due to the fact that the

| Dimension | Basic prompt | Augmented prompt |
|---|---|---|
| Style | 4.33 | 4.10 |
| Clarity | 4.34 | 4.24 |
| Correctness | 4.43 | 4.29 |
| Tone | 3.34 | 3.59 |
| Accuracy | 3.77 | 3.92 |
| Specificity | 3.29 | 3.46 |
| Completeness | 2.72 | 2.97 |
| Uncertainty | 2.73 | 2.92 |

Table 3: Comparing GPT-4's performance with and without the inclusion of a description of the evaluation criteria in the prompt.

experiments were run almost two months apart, and the latest were carried out after a major new release[9].

Interestingly, the addition of the evaluation criteria helps the LLM provide better answers for the most difficult presentation dimension, Tone, and all the epistemological ones. We notice also that the improvement on the difficult dimensions comes at the cost of degraded performance on the presentational dimensions, supporting the idea that there may be unavoidable trade-offs between surface and epistemological content quality. This direction needs further research and experimentation but already seems promising.

## B  ARGUMENTATIVE PATTERNS

What is typical of LLMs' answers, and how may gold standard answers differ, if they were available? While we cannot fully answer this question yet, the findings from our evaluation provide some additional insights. One interesting aspect emerges with respect to the argumentative structure of LLMs' answers. We use two examples from one of the models tested, to ground the discussion, but similar arguments hold for all models.

**Example 1.** LLMs often spend a substantial part of the answer on generic, cautionary, sometimes preachy, reminders; e.g., see the highlighted text in the example below:

| **Question** | Is global warming caused by the Pacific decadal oscillation? |
|---|---|
| **LLM** | No, global warming is not caused by the Pacific Decadal Oscillation (PDO). The PDO is a long-term ocean temperature fluctuation in the Pacific Ocean that shifts between warm and cool phases approximately every 20 to 30 years. While these shifts can influence regional climate patterns and marine ecosystems, they do not cause global warming. Global warming is primarily caused by human activities, particularly the burning of fossil fuels like coal, oil, and gas, which increase the concentration of greenhouse gases in the atmosphere and trap heat. |

It is not clear whether the highlighted part contributes to answering the question for all audiences. For example, the result may be sub-optimal *cooperative conversation*, in a Gricean's sense (Levinson, 1983). Specifically, one could argue the *maxim of quantity* is being violated (do not provide more information than required). A person who is asking a very specific question about climate change can be assumed to already be familiar with its basic mechanism. Another maxim that may be violated is the *maxim of manner* (be relevant): the question asks specifically about 'decadal oscillation', so the answer should be specific to that. In addition, the presentation style may not appeal to the different audiences of science communication and possibly lead to diminished interest and fatigue (Schäfer et al., 2018).

---

[9]The augmented prompt experiment, and control, were performed after the OpenAI announcements on Nov 6, cf. `https://openai.com/blog/new-models-and-developer-products-announced-at-devday`.

The provided answers are different to similar questions by specialized websites that list carefully compiled human answers.[10] These answers tend to rely very much on direct and specific evidence from multiple sources such as land, atmospheric and ocean data for temperature trends.[11]

**Example 2.** A related argumentative pattern can be also observed frequently, illustrated by the example below:

| | |
|---|---|
| **Question** | Are we heading towards an ice age? |
| **LLM** | No, we are not heading towards an ice age. The current scientific consensus is that human activities, particularly the emission of greenhouse gases, are causing the planet to warm at an unprecedented rate. This rapid warming is leading to a variety of climate changes, including more frequent and severe heatwaves, rising sea levels, and changes in precipitation patterns. While natural processes have triggered ice ages in the past, these are very slow processes that take thousands of years, and the current rapid warming trend is not consistent with the onset of a new ice age. |

Similar observations could be made, as in the previous example, regarding the pragmatics aspects of *effective conversation*, for the choices of arguments made in the answer. But the answer above relies crucially, although generically, on the notion of *scientific consensus*; which happens relatively frequently in our data. Besides the possibility of being superficially interpreted as relying primarily on 'arguments from authority', research suggests that the 'consensus' argument can be surprisingly ineffective in some cases. Orchinik et al. (2023) show that there is a complex belief system underlying how such arguments are interpreted. This depends, among other factors, on how scientists are perceived in terms of credibility and skills. Orchinik et al. (2023) argue that perceived credibility, which in turn may depend on general worldview, affects how consensus-based messages are received and receptiveness to future messaging. From this perspective, addressing some audiences, simple consensus messaging may be not only sub-optimal from a Gricean perspective, but also ineffective.

We do not know how the current style of presentation and argumentation emerges in LLMs, but the LLMs we study are similar in this respect. Our framework captures these aspects in the *tone* and *specificity* dimensions, but one should consider assessing this directly in the future.

## C  QUESTIONS

In this section we explain the pipeline used for selection, generation, post-processing and sampling climate change related questions. The question set consists of 300 questions, with 100 questions gathered from 3 sources each: i) Synthetic questions generated based on Wikipedia articles, ii) Manually rephrased questions based on Skeptical Science website, and iii) questions taken from Google Trends.

### C.1  SYNTHETIC QUESTIONS FROM WIKIPEDIA

We started by gathering a set of Wikipedia articles related to climate change. We followed 3 strategies to select climate related articles from Wikipedia. Following the first strategy (REF.), we gather all the Wikipedia articles that are referenced in the main "Climate Change" article.[12] In the second strategy (CAT.), we select all the articles that are directly listed in the climate change category. Finally, to cover regional articles (REG.), we manually curate a list of articles with titles *"Climate Change in [country/region]"*. From a pool of articles gathered following these 3 strategies, we selected paragraphs within an article if the paragraph consists of more than 500 characters. In total, we obtained 1969 paragraphs from Wikipedia. The following table reports a break-down of number of paragraphs based on the selection strategy:

---

[10]E.g., https://climatefeedback.org/ or https://skepticalscience.com/.

[11]They are also heavily backed by visual quantitative data.

[12]https://en.wikipedia.org/wiki/Climate_change

| Strategy | # Articles | # Paragraphs |
|----------|-----------|--------------|
| REF. | 35 | 858 |
| CAT. | 46 | 434 |
| REG. | 48 | 677 |
| Total | 129 | 1969 |

We then input each selected paragraph in `GPT-4`. We ask the model to generate as many questions as possible that can be answered using the paragraph. The model is instructed to only generate questions that are salient and related to climate change. This process resulted in 15265 questions. We post process the questions and remove undesirable ones with 4 filters that we explain next.

**Climate Change Filter.** We remove all questions that are not climate change related. We use the climate-bert (Webersinke et al., 2022) classifier and label each question with two labels: climate related and not climate related. We remove 2647 questions that are not classified as climate-related questions.

**Duplicate Filter.** We remove questions that are a duplicate of another question. To this end, we embed all questions using a universal sentence encoder (Yang et al., 2020).[13] We consider two questions as duplicates if the cosine similarity between their embeddings is greater than $0.85$. Therefore, we remove 1188 questions that are duplicates of other questions.

**Context Dependent Filter.** We filter out questions that are taken out of context. The reason that this filter is necessary is that we generate questions from paragraphs, therefore, some questions are nonsensical when they are not accompanied by the corresponding Wikipedia paragraph. An example of such a question is: *"What are the two classes of climate engineering discussed in the study?"*; without knowing which study is referred to, this question cannot be answered. To develop this filter, we build a dedicated classifier. Specifically, we manually annotate 100 questions with two labels: context dependent, and not context dependent. Next, contextualize the question with the instruction *"Write Yes if the query is taken out of context, write No otherwise."* and extract the last layer's representations of a `flan-xxl` encoder (Chung et al., 2022). Finally, we train a logistic regression classifier on the representations to detect context dependent questions. We find the context dependency filter to be $97\%$ accurate on 100 manually annotated validation questions. Using this classifier, we detect 552 context dependent questions.

**Specificity Filter.** We remove questions that are asking about a very specific and narrow topic. In our study, we aim to evaluate large language models on a set of challenging and multifaceted questions that target information needs of users related to climate change. Therefore, questions that ask for a specific detail are not the target of this study and are typically easy to answer. An example of such question is: "What was the reason for shutting down reactor number one of the Fessenheim Nuclear Power Plant on 4 August 2018?" To remove such specific questions, we again build a light-weight logistic regression classifier on top of `flan-xxl` representations. We contextualize each question with the instruction: *"Write Yes if the following query is asking about a specific subject, write No otherwise"*. We then extract the contextualized representations from the last layer of `flan-xxl` and feed that to a logistic regressor. We find the specificity filter to be $84\%$ accurate on a sample of 100 annotated validation questions. We detect and remove 5472 specific questions.

After applying all 4 filters, the final post-processed question set consists of 5404 questions. The question set that is rated in our evaluation framework consists of 100 questions from each source. This means that we need to sample 100 diverse questions from this pool of $\approx$ 5k questions. To make sure that we cover different topics and type of questions, we first label each question with the topic and properties of the question, and then sample a 100 validation questions, where different topics and properties are equally presented. Next, we explain the classifiers that are developed for labeling the questions.

**Topic Classifier.** We use the approach as above and train a logistic regression classifier on top of `flan-xxl` encoder to classify questions based on the topics. Inspired by IPCC chapters, we con-

---

[13]We use `universal-sentence-encoder-qa/3` model.

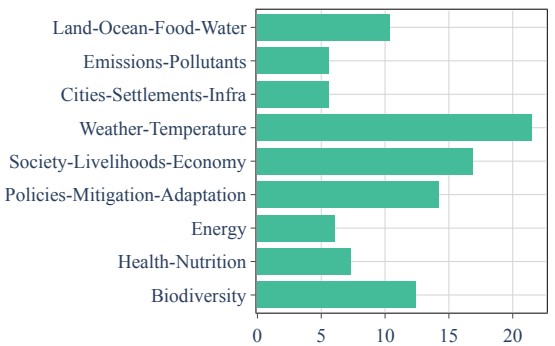 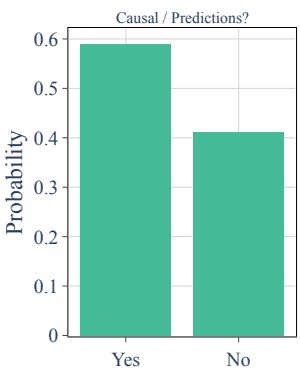

Figure 5: Left: Distribution of the topic of the synthetic questions. The x axis shows the percentage of questions in each topic. The most frequent topic is "Weather-Temperature" topic. Right: Distribution of the causal feature of synthetic questions. There are more questions that are asking causal questions, or questions about predicting the future compare to the rest of the questions.

sider the following 9 topics: *"Energy", "Emissions-Pollutants", "Policies-Mitigation-Adaptation", "Weather-Temperature", "Land-Ocean-Food-Water", "Society-Livelihoods-Economy", "Health-Nutrition", "Biodiversity", "Cities-Settlements-Infra"*. We find this classifier to be 75% accurate on a sample of 100 annotated validation questions. The distribution of predicted questions' topics is depicted in Figure 5.

**Causal Prediction Classifier.** An important and challenging type of questions that one can ask about climate change is about causes or effects of climate change, or predictions about the future. To detect this type of questions, we classify questions into two classes: causal-prediction class and others. The instruction that is used for contextualizing the questions is: *"Write Yes if the following query is asking about causes or effects of something, or is asking about predictions about the future. write No otherwise"*. We find this classifier to be 93% accurate on a sample of 100 annotated validation questions. The distribution of predictions is shown in Figure 5

For synthetic Wikipedia questions, we sample 108 questions, 6 from each of the 18 possible combinations of topic and causal-prediction features. We sample an equal number of questions from the other two sources, Skeptical Science and GTrends. We manually inspect all questions from all the 3 different sources. The final question set used in the evaluations consists of 300 questions, i.e., 100 questions from each source.

# D  PROMPTS

Please refer to Table 4 for an extensive list of prompts used to generate the data used throughout the paper.

# E  ANSWER STATISTICS

We report the average number of sentences and the average number of words per sentence for all models evaluated in Table 5. Although in the prompts used for answer generation we explicitly instruct the model to only use 3 to 4 sentences to answer the question, we observe that most models generate between 4 to 5 sentences. Furthermore, InstructGPT (turbo), GPT-4, and Falcon-180B-Chat, generate longer sentences compared to the other models.

---

[14]We found that we have used *"an statement"* instead of *"a statement"* in our experiments. We did not rerun the experiments as we believe that LLMs are quite robust to minor typos and results should not be significantly affected by it. Also, doing otherwise would be wasteful.

| Function | Prompt |
|---|---|
| Answer Generation | You are an expert on climate change communication. Answer each question in a 3-4 sentence paragraph. |
| Obtain URL | Please provide a Wikipedia article that supports your answer. Just state the url, do not include additional text. If there is no Wikipedia url supporting the answer just say "No URL". |
| Extract Keypoints | Now go through all the statements made in the answer. Mention 1 to 3 key statements that are made to answer the question. If you can not provide key statement/statements, only write No Keypoints. It is very important to copy the statements verbatim from the answer. |
| Rate Passages | You are given a statement[14] and a passage from Wikipedia. Rate how useful the passage is for evaluating the statement on a scale from 0 (completely irrelevant) to 100 (supports or contradicts the statement). Rate the passage high only if it supports or contradicts the statement. Just state the numbers in one line, nothing else. Statement: [keypoint] Passage: [par] |
| Presentational AI Assistance | Given the following question and answer, express your disagreement with the statement in a concise sentence in a single line. You may be provided with relevant paragraphs from Wikipedia, if so, you must use those verbatim to support your critique. If you fully agree with the statement, state "No Critique". Question: [question] Answer: [answer] Statement: [statement] |
| Style Statement | The information is presented well for a general audience. In particular, the answer is not too long or too short, there is no repetition in the text, and the answer is not too informal or too technical. |
| Clarity Statement | The answer is clear and easy to understand. For example, if there are numbers and formulae in the answer, they are easy to understand. Furthermore, sentences are not too long or too short. |
| Correctness Statement | The language in the answer does not contain mistakes. In particular, there are no grammatical, spelling, or punctuation errors. |
| Tone Statement | The tone of the answer is neutral and unbiased. In particular, the tone is not negative and the answer does not try to convince the reader of an opinion or belief. |
| Epistemological AI Assistance | Given the following question and answer, express your disagreement with the statement in a concise sentence in a single line. You may be provided with relevant paragraphs from Wikipedia, if so, you must use those verbatim to support your critique. If you fully agree with the statement, state "No Critique". Question: [question] Answer: [answer] Statement: [statement]. |
| Accuracy Statement | The answer is accurate. In particular, it does not take scientific findings out of context, does not contradict itself, does not rely on anecdotal evidence, and does not misuse key terms or scientific terminology. |
| Specificity Statement | There is no irrelevant statement with respect to the question in the answer, and there is no vague or generic statement in the answer. |
| Completeness Statement | The answer addresses everything the question asks for. In particular, it does not miss any part of the question and provides enough necessary details, e.g., numbers, statistics, and details. If the question asks for a specific time range or region, the answer correctly provides that information. |
| Uncertainty Statement | If there is an uncertainty involved in the scientific community, the answer appropriately conveys that uncertainty. Note that it may be appropriate not to mention uncertainty at all. |

Table 4: Prompts used to generate answers, AI Assistance and evidence.

| | InstructGPT | | | ChatGPT | PaLM-2 | GPT4 | Falcon |
| | davinci-002 | davinci-003 | turbo | | text-bison | | 180B-Chat |
|---|---|---|---|---|---|---|---|
| # Sentences | 4.99 | 3.11 | 3.42 | 4.07 | 4.47 | 4.33 | 3.81 |
| # Words per sentence | 14.3 | 18.68 | 21.49 | 20.66 | 19.67 | 21.52 | 22.03 |

Table 5: Average number of sentences and words per sentence for each model. We observe 4 out of 7 models generate 4 to 5 sentences, and `Falcon-180B-Chat` generates longer sentences compared to the other models in the batch.

| | Per Example | | Per Keypoint | |
| | Percentage % | Count | Percentage % | Count |
|---|---|---|---|---|
| Fully Supports | 6.95 | 16 | 12 | 124 |
| Partially Supports | 39.13 | 90 | 54.79 | 566 |
| No Support | 53.91 | 124 | 32.81 | 339 |
| Contradicts | 0 | 0 | 0.38 | 4 |
| Total | 100 | 230 | 100 | 1033 |

Table 6: AIS results for `GPT-4` generations. We report the results both at example level, where an example can consist of multiple keypoints, and at keypoint level. We observe that the majority of keypoints are only partially supported, and only around 7% of examples are fully supported.

## F  AIS EVALUATION

In our AIS evaluation we ask human raters to assess whether each keypoint is, fully, partially, or not supported by its corresponding top 3 retrieved passages. At the keypoint level, as shown in Figure 4, the majority of keypoints are partially supported, while only 12% of keypoints are fully supported by their corresponding evidence. We further ask the raters whether the top 3 passages jointly provide support for the keypoint. We then aggregate this result in the following way. We consider an answer to be fully attributed if all its keypoints are supported. An answer is not supported if all its keypoints are not supported. Otherwise, the answer is partially supported. An example is given in Table 7. We observe (Figure 4) that only 6.95% of answers are fully supported.

## G  VALIDATION EXPERIMENTS

To validate our evaluation framework, we manually created a set of 30 questions and answers, each exhibiting a specific issue from our full list of issues. We pose this as a task to our raters in two settings: with and without AI Assistance, each replicated three times.

We found (cf. Table 9) that without assistance, most of the issues (83%) were found by at least one rater, but only 60% of issues by the majority of raters.

With the help of assistance, more issues were detected (93.33% were detected by at least one rater, 76.67% by the majority of raters). There was just one issue not detected by any rater: *anecdotal evidence*[15].

## H  INTER-RATER AGREEMENT

We first measure the agreement among raters when rating each dimension on the likert scale. In particular, we report two metrics of agreement:

**Pairwise distance.** We measure the average pairwise distance between the ratings. More specifically, for any 2 raters (out of 3 raters) rating the same example, we compute the absolute distance

---

[15]All raters did correctly rate the dimension ('accuracy' as 1 or 2), but highlighted only the 'incorrect' issue. The answer was incorrect, based on anecdotal evidence.

| Question | How could pests and diseases spread more easily under warmer conditions caused by climate change? |
|---|---|
| **Answer** | Climate change, particularly global warming, can facilitate the spread of pests and diseases more easily. Warmer conditions can extend the breeding seasons of pests, increase their population growth, and expand their geographical range. For instance, mosquitoes, which are vectors for diseases like malaria and dengue, thrive in warmer climates and can reproduce and mature more quickly under such conditions. Additionally, warmer temperatures can also accelerate the lifecycle of pathogens, making diseases more virulent and their transmission more efficient. |
| **1st Keypoint** | Warmer conditions can extend the breeding seasons of pests, increase their population growth, and expand their geographical range. |
| **Pragraph 1.** | Infectious diseases that are sensitive to climate can be grouped into: vector-borne diseases (transmitted via mosquitos, ticks etc.), water-borne diseases (transmitted via viruses or bacteria, e.g.: E. Coli) and food-borne diseases (e.g.: Salmonella). Climate change is affecting the distribution of these diseases due to the expanding geographic range and seasonality of these diseases and their vectors. |
| **Pragraph 2.** | Mosquito-borne diseases that are sensitive to climate include malaria, elephantiasis, Rift Valley fever, yellow fever, dengue fever, Zika virus, and chikungunya. Scientists found in 2022 that rising temperatures are increasing the areas where dengue fever, malaria and other mosquito-carried diseases are able to spread. Warmer temperatures are also advancing to higher elevations, allowing mosquitoes to survive in places that were previously inhospitable to them. This risks malaria making a return to areas where it was previously eradicated. |
| **Pragraph 3.** | Ticks are changing their geographic range because of rising temperatures, and this puts new populations at risk. Ticks can spread lyme disease and tick-borne encephalitis. It is expected that climate change will increase the incidence of these diseases in the Northern Hemisphere. For example, a review of the literature found that "In the USA, a 2°C warming could increase the number of Lyme disease cases by over 20% over the coming decades and lead to an earlier onset and longer length of the annual Lyme disease season". |
| **Rating** | Not Supported |
| **2nd Keypoint** | Mosquitoes, which are vectors for diseases like malaria and dengue, thrive in warmer climates and can reproduce and mature more quickly under such conditions. |
| **Paragraphs** | . . . |
| **Rating** | Supported |
| **3rd Keypoint** | Warmer temperatures can also accelerate the lifecycle of pathogens, making diseases more virulent and their transmission more efficient. |
| **Paragraphs** | . . . |
| **Rating** | Supported |
| **Overall Rating** | Partially Supported |

Table 7: An example of the AIS task with ratings. Overall, the answer is only partially supported, as the first keypoint is not supported by the evidence.

| **Question.** When did the general public become aware of carbon dioxide's effects on climate? | |
|---|---|
| **Answer.** The general public became significantly aware of carbon dioxide's effects on climate change around the late 20th century. The Intergovernmental Panel on Climate Change (IPCC) was established in 1988, and its assessments, which highlighted the role of carbon dioxide in global warming, were widely disseminated and discussed in the media. However, it's important to note that the scientific understanding of carbon dioxide's impact on climate predates this public awareness by many decades, with initial theories and evidence emerging as early as the late 19th century. | **AI Assistance.** While the answer provides a general timeline of public awareness of carbon dioxide's effects on climate change, it fails to mention key milestones such as the warnings of the climate crisis by Glenn T. Seaborg in 1966, or John Sawyer's study in 1972, which summarized the knowledge of the science at the time and accurately predicted the rate of global warming for the period between 1972 and 2000. These events significantly contributed to the public's understanding of the issue before the establishment of the IPCC in 1988. |
| **AIS.** `Fully attributable` | **Average completeness rating.** `1.6` |

| **Question.** How do elevated carbon dioxide levels affect the mineral density of crops and wild plants? | |
|---|---|
| **Answer.** Elevated carbon dioxide levels can significantly affect the mineral density of crops and wild plants. While higher $CO_2$ levels can stimulate plant growth, they can also reduce the concentration of essential minerals in a variety of crops and wild plants. This phenomenon, known as "$CO_2$ dilution effect", results in crops with lower nutritional value, including reduced levels of protein, iron, zinc, and other essential nutrients. Therefore, while plants may grow larger in response to increased $CO_2$, their nutritional quality can be significantly compromised. | **AI Assistance.** The statement fails to acknowledge the uncertainty in the potential positive impacts of elevated $CO_2$ levels, such as higher crop yields due to $CO_2$ fertilization, and the possibility of more arable land becoming available as frozen land thaws. |
| **AIS.** `Fully attributable` | **Average uncertainty rating.** `1.6` |

| **Question.** Can climate sensitivity be ignored because too low? |
|---|
| **Answer.** Climate sensitivity, which refers to the temperature change associated with a doubling of the concentration of carbon dioxide in the atmosphere, cannot be ignored even if it is low. This is because even a small increase in global temperatures can have significant impacts on the Earth's climate system. For instance, it can lead to more frequent and severe weather events, such as hurricanes and droughts, and cause sea levels to rise, which can result in coastal flooding. Therefore, understanding and addressing climate sensitivity is crucial in climate change mitigation and adaptation strategies. |
| **AIS.** `Not Supported`             **Average accuracy rating.** `4.7` |

| **Question.** Is global warming caused by the pacific decadal oscillation? |
|---|
| **Answer.** No, global warming is not caused by the Pacific Decadal Oscillation (PDO). The PDO is a long-term ocean temperature fluctuation in the Pacific Ocean that shifts between warm and cool phases approximately every 20 to 30 years. While these shifts can influence regional climate patterns and marine ecosystems, they do not cause global warming. Global warming is primarily caused by human activities, particularly the burning of fossil fuels like coal, oil, and gas, which increase the concentration of greenhouse gases in the atmosphere and trap heat. |
| **AIS.** `Not Supported`             **Average specificity rating.** `5` |

Table 8: Examples highlighting the differences between attribution scores and epistemological ratings under our framework.

|          | Without AI Assistance (3x) | With AI assistance (3x) |
|----------|----------------------------|-------------------------|
| Any      | 83.33 %                    | 93.33 %                 |
| Majority | 60.00 %                    | 76.67 %                 |
| All      | 33.33 %                    | 43.33 %                 |

Table 9: Validation results. The percentage of the (30) issues recognized by any rater, the majority of raters or all of the raters.

between the values they chose from the likert scale[16] and report the average for each dimension in Table 11. In general, we observe a reasonably high agreement among the raters, as the average distance is close to or below 1 in most dimensions. Notably, we observe a higher agreement in the presentational dimensions *style*, *clarity*, and *correctness*.

**Krippendorff's alpha.** In addition to pairwise distances, we compute Krippendorff's alpha. Krippendorff's alpha measures $1 - \frac{D_o}{D_e}$, where $D_o$ is the observed disagreement, and $D_e$ is the expected disagreement by chance. Values are in $[-1, 1]$ range, where 1 means complete agreement and $-1$ means complete systematic disagreement. Numbers in Table 12 suggest a similar trend to pairwise distance, where in most dimensions the agreement is medium to high, and the agreement in most presentational dimensions is higher compared to epistemological dimensions.

Furthermore, we measure the agreement among raters when choosing issues. A rater might select or not select a given issue for a given answer, therefore, the value of interest is a binary variable. As above report two metrics of agreement:

**Pairwise agreement.** We look at the agreement among raters when selecting or not selecting a given issue. Particularly, we consider 2 raters to agree with each other on a certain issue for a given answer if they both select or both not select that issue. We then report the percentage of pairwise agreement per issue in Table 13. For the majority of issues we observe a high agreement among raters. As one might expect, issues such as "not enough detail", "vague", "uncertainty missing", and "biased" are more controversial and we see a lower agreement among the raters.

**Krippendorff's alpha.** Similarly, we compute the Krippendorff's alpha for agreement on issues and observe a similar trend in Table 14.

Looking at Table 10 we note that some issues are rarely chosen by raters and thus pairwise agreement numbers might be artificially high. For a deeper understanding regarding how well raters are able to agree on a specific issue we compute Krippendorff's alpha only for low ratings, i.e. cases where raters are required to select one or more issues. We report these numbers for a subset of dimensions with higher incidence counts in Table 15. As hinted by incidence prevalence in Table 10, we find that when raters agree on a low rating for an epistemological dimension, they also exhibit medium to high agreement on what the specific issue is. One exception is *accuracy:incorrect* which might be too generic as an issue.

Overall, agreement on specific issues is not high enough to recommend our 3-rater setup for evaluation of individual answers but for comparing and highlighting the strengths and shortcomings of models on a system level, as indicated by the fairly tight error bars in Figure 2.

## I   BREAKDOWN OF RATINGS PER QUESTION TYPE

We compare the presentational and epistemological adequacy of GPT-4 answers, based on the question source, type, and causal-prediction dimension, as described in Appendix C.1. Generally, there isn't a significant difference between the ratings based on the topic of the question as shown in Figure 6. However, we observe that questions in the *"Policies-Mitigation-Adaptation"* category receive lower ratings in most of the epistemological dimensions, and particularly in the *tone* dimension. We further look at the difference in average ratings based on the source of the question (Wikipedia, Skeptical Science, or GTrends), and causality of the question. The source of the question does not

---

[16]In our interface the raters agree with a statement (see Table 19) on a 5-point scale between *disagree completely* to *neither* to *agree completely* which we map to $1 \ldots 5$. See Figure 15 for a screenshot.

| Issue | InstructGPT | | | ChatGPT | PaLM-2 | GPT4 | Falcon |
| --- | --- | --- | --- | --- | --- | --- | --- |
| | davinci-002 | davinci-003 | turbo | | text-bison | | 180B-Chat |
| **style** | | | | | | | |
| inconsistent | 4.88 | 1.00 | 0.33 | 0.00 | 1.11 | 0.22 | 0.45 |
| repetitive | 20.15 | 3.11 | 0.11 | 0.56 | 1.45 | 1.11 | 0.33 |
| too informal | 4.11 | 1.11 | 0.22 | 0.11 | 1.78 | 1.44 | 0.89 |
| too long | 1.03 | 1.67 | 0.33 | 0.89 | 2.12 | 2.11 | 0.89 |
| too short | 10.14 | 8.56 | 0.22 | 0.22 | 2.56 | 0.33 | 1.11 |
| other | 2.95 | 1.00 | 0.22 | 0.00 | 0.78 | 0.67 | 0.45 |
| **clarity** | | | | | | | |
| hard math | 1.67 | 0.44 | 1.67 | 0.33 | 0.67 | 1.56 | 0.00 |
| sentences too long | 1.80 | 1.33 | 0.11 | 0.22 | 1.67 | 3.11 | 1.22 |
| too technical | 3.59 | 1.00 | 0.33 | 0.44 | 1.22 | 2.56 | 0.56 |
| other | 8.60 | 1.00 | 0.33 | 0.11 | 1.56 | 0.44 | 0.78 |
| **correctness** | | | | | | | |
| incomplete sentence | 3.47 | 2.44 | 0.78 | 0.00 | 0.00 | 0.00 | 0.00 |
| incorrect grammar | 6.29 | 0.33 | 0.33 | 0.11 | 0.11 | 0.11 | 0.67 |
| incorrect punctuation | 2.18 | 0.44 | 0.56 | 0.00 | 0.11 | 0.22 | 0.56 |
| incorrect spelling | 0.77 | 0.00 | 0.11 | 0.11 | 0.22 | 0.00 | 0.11 |
| other | 3.98 | 1.11 | 0.00 | 0.11 | 0.45 | 0.11 | 0.78 |
| **tone** | | | | | | | |
| biased | 28.50 | 34.44 | 24.78 | 42.38 | 33.85 | 30.33 | 23.72 |
| negative | 1.28 | 1.78 | 1.56 | 1.00 | 5.01 | 3.00 | 1.89 |
| persuasive | 2.57 | 8.00 | 4.80 | 7.68 | 10.69 | 8.00 | 4.45 |
| other | 0.39 | 0.67 | 0.22 | 0.11 | 0.45 | 2.00 | 0.22 |
| **accuracy** | | | | | | | |
| anecdotal | 10.78 | 1.33 | 3.35 | 19.24 | 5.90 | 2.56 | 3.01 |
| incorrect | 20.92 | 10.78 | 5.58 | 3.23 | 11.69 | 4.44 | 4.57 |
| science out of context | 9.37 | 6.11 | 5.69 | 2.67 | 5.35 | 3.78 | 2.56 |
| self contradictory | 2.70 | 0.89 | 0.11 | 0.11 | 0.89 | 0.44 | 0.33 |
| wrong use of terms | 1.93 | 1.22 | 1.00 | 0.33 | 1.45 | 0.44 | 0.22 |
| other | 3.34 | 3.00 | 1.90 | 1.89 | 2.00 | 5.67 | 0.78 |
| **specificity** | | | | | | | |
| irrelevant info | 15.15 | 4.56 | 3.79 | 5.12 | 8.69 | 8.89 | 4.01 |
| vague | 49.42 | 44.78 | 48.88 | 58.40 | 51.67 | 39.11 | 35.86 |
| other | 1.67 | 3.44 | 1.45 | 0.56 | 2.12 | 1.67 | 1.34 |
| **completeness** | | | | | | | |
| does not address main parts | 29.91 | 22.56 | 11.16 | 9.79 | 15.92 | 8.78 | 9.47 |
| does not address region | 3.34 | 2.67 | 0.78 | 0.56 | 1.34 | 1.78 | 1.22 |
| does not address time | 2.05 | 4.11 | 1.90 | 0.67 | 0.67 | 2.78 | 0.67 |
| ignores science | 9.11 | 14.11 | 6.92 | 5.01 | 10.47 | 5.44 | 3.01 |
| not enough detail | 52.89 | 60.22 | 64.06 | 79.53 | 58.13 | 61.22 | 51.89 |
| other | 1.16 | 0.89 | 0.45 | 0.11 | 0.89 | 2.78 | 1.11 |
| **uncertainty** | | | | | | | |
| consensus missing | 19.77 | 14.89 | 21.99 | 9.34 | 12.14 | 9.89 | 9.80 |
| contradicting evidence missing | 4.11 | 6.33 | 2.57 | 2.00 | 4.23 | 3.56 | 2.90 |
| uncertainty missing | 57.25 | 75.00 | 72.88 | 87.65 | 71.94 | 76.78 | 58.02 |
| other | 0.90 | 1.11 | 0.45 | 0.11 | 0.45 | 1.89 | 0.33 |

Table 10: Percentage of specific issues identified by raters.

| Issue | InstructGPT | | | ChatGPT | PaLM-2 | GPT4 | Falcon |
| | davinci-002 | davinci-003 | turbo | | text-bison | | 180B-Chat |
|---|---|---|---|---|---|---|---|
| style | 1.12 | 0.95 | 0.76 | 0.61 | 0.88 | 0.79 | 0.75 |
| clarity | 0.97 | 0.74 | 0.73 | 0.59 | 0.69 | 0.81 | 0.69 |
| correctness | 0.98 | 0.69 | 0.66 | 0.56 | 0.59 | 0.62 | 0.68 |
| tone | 1.16 | 1.26 | 1.21 | 1.30 | 1.36 | 1.22 | 1.23 |
| accuracy | 1.05 | 0.97 | 1.07 | 1.15 | 1.13 | 0.97 | 0.95 |
| specificity | 1.04 | 1.16 | 1.06 | 0.98 | 1.23 | 1.26 | 1.20 |
| completeness | 1.00 | 1.03 | 1.06 | 0.71 | 1.13 | 1.01 | 1.21 |
| uncertainty | 0.95 | 0.98 | 0.89 | 0.57 | 1.10 | 0.78 | 1.26 |

Table 11: Average pairwise distance between likert ratings for each dimension. Distances between ratings on presentational adequacy are generally lower compared to epistemological adequacy.

| Issue | InstructGPT | | | ChatGPT | PaLM-2 | GPT4 | Falcon |
| | davinci-002 | davinci-003 | turbo | | text-bison | | 180B-Chat |
|---|---|---|---|---|---|---|---|
| style | 0.45 | 0.53 | 0.74 | 0.70 | 0.60 | 0.48 | 0.72 |
| clarity | 0.59 | 0.73 | 0.60 | 0.72 | 0.72 | 0.65 | 0.77 |
| correctness | 0.57 | 0.74 | 0.80 | 0.85 | 0.82 | 0.71 | 0.78 |
| tone | 0.48 | 0.36 | 0.41 | 0.31 | 0.25 | 0.36 | 0.41 |
| accuracy | 0.56 | 0.57 | 0.52 | 0.46 | 0.46 | 0.59 | 0.62 |
| specificity | 0.53 | 0.40 | 0.50 | 0.51 | 0.32 | 0.32 | 0.39 |
| completeness | 0.57 | 0.48 | 0.47 | 0.64 | 0.38 | 0.46 | 0.37 |
| uncertainty | 0.59 | 0.51 | 0.57 | 0.75 | 0.40 | 0.63 | 0.32 |

Table 12: Krippendorff's alpha of 3 likert ratings per dimension. In general we observe a medium agreement. For most LLMs the value is higher for the presentational dimensions, except tone.

affect the ratings significantly (please refer to Figure 7). However, we observe that Wikipedia questions tend to receive lower epistemological adequacy ratings. This could be because these questions ask for more details and very specific info compared to GTrends and Skeptical Sciences, and thus are harder to answer.

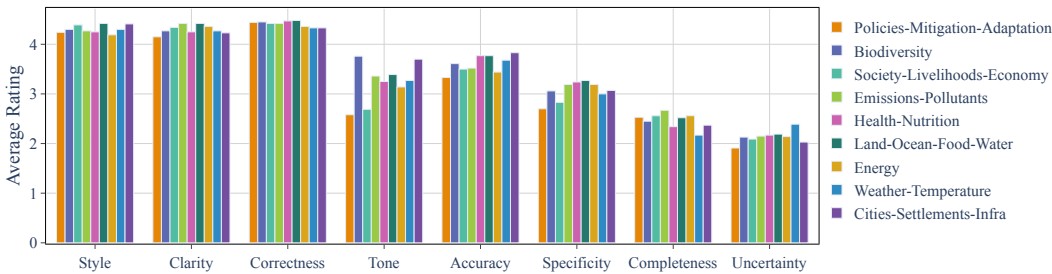

Figure 6: Average rating along all the dimensions per question topic. Questions in the *"Policies-Mitigation-Adaption"* category receive lower ratings in most of the epistemological dimensions, and particularly in "Tone" dimension.

## J  TIMING ANALYSIS

We analyze how long raters take for their tasks. As can be seen in Figure 8, rating the epistemological dimensions generally takes more time than assessing the presentation quality, even though the latter is done first in our questionnaire. We also observe that for most systems the screening part, which includes the initial reading of question and answer, takes longer than rating the presentational

| Issue | InstructGPT | | | ChatGPT | PaLM-2 | GPT4 | Falcon |
|---|---|---|---|---|---|---|---|
| | davinci-002 | davinci-003 | turbo | | text-bison | | 180B-Chat |
| **style** | | | | | | | |
| too informal | 92.40 | 97.77 | 99.55 | 99.78 | 96.42 | 97.10 | 98.21 |
| too long | 98.20 | 98.00 | 99.33 | 98.44 | 96.42 | 95.77 | 98.44 |
| too short | 84.79 | 87.42 | 99.66 | 99.55 | 95.08 | 99.33 | 97.77 |
| inconsistent | 90.72 | 98.00 | 99.33 | 100.00 | 97.76 | 99.55 | 99.11 |
| repetitive | 83.63 | 96.88 | 99.78 | 98.88 | 97.76 | 97.77 | 99.33 |
| other | 94.33 | 98.00 | 99.55 | 100.00 | 98.88 | 98.66 | 99.33 |
| **clarity** | | | | | | | |
| sentences too long | 96.39 | 97.77 | 99.78 | 99.55 | 96.64 | 94.21 | 97.54 |
| too technical | 94.07 | 98.22 | 99.33 | 99.11 | 97.76 | 95.10 | 98.88 |
| hard math | 96.91 | 99.11 | 97.31 | 99.33 | 98.66 | 96.88 | 100.00 |
| other | 85.95 | 98.22 | 99.55 | 99.78 | 97.76 | 99.11 | 98.44 |
| **correctness** | | | | | | | |
| incomplete sentence | 94.33 | 97.11 | 99.55 | 100.00 | 100.00 | 100.00 | 100.00 |
| incorrect spelling | 98.45 | 100.00 | 99.78 | 99.78 | 99.55 | 100.00 | 99.78 |
| incorrect punctuation | 95.88 | 99.11 | 98.88 | 100.00 | 99.78 | 99.55 | 98.88 |
| incorrect grammar | 89.43 | 99.33 | 99.33 | 99.78 | 99.78 | 99.78 | 98.66 |
| other | 93.81 | 98.22 | 100.00 | 99.78 | 99.33 | 99.78 | 98.66 |
| **tone** | | | | | | | |
| biased | 60.57 | 59.02 | 67.15 | 48.33 | 57.06 | 59.19 | 64.06 |
| persuasive | 95.62 | 87.08 | 91.03 | 86.38 | 81.05 | 84.98 | 91.96 |
| negative | 97.68 | 96.66 | 97.09 | 98.21 | 90.92 | 95.07 | 96.65 |
| other | 99.23 | 98.66 | 99.55 | 99.78 | 99.33 | 96.08 | 99.55 |
| **accuracy** | | | | | | | |
| incorrect | 69.91 | 89.73 | 91.43 | 95.70 | 84.36 | 92.20 | 93.02 |
| science out of context | 82.70 | 89.35 | 88.76 | 95.22 | 89.17 | 92.46 | 95.56 |
| self contradictory | 95.49 | 98.20 | 99.87 | 99.76 | 98.40 | 98.98 | 99.24 |
| anecdotal | 78.05 | 97.18 | 92.37 | 63.80 | 87.17 | 94.63 | 94.67 |
| wrong use of terms | 96.22 | 97.69 | 98.26 | 99.52 | 97.46 | 98.98 | 99.49 |
| other | 93.46 | 93.84 | 96.12 | 96.42 | 95.45 | 89.13 | 98.48 |
| **specificity** | | | | | | | |
| irrelevant info | 75.59 | 90.79 | 92.43 | 89.84 | 84.89 | 84.06 | 93.60 |
| vague | 48.96 | 54.20 | 60.44 | 58.90 | 52.60 | 56.81 | 60.66 |
| other | 97.13 | 93.94 | 97.71 | 99.09 | 96.31 | 97.00 | 97.27 |
| **completeness** | | | | | | | |
| does not address main parts | 61.33 | 69.27 | 80.68 | 82.77 | 75.12 | 84.07 | 84.95 |
| does not address region | 93.36 | 94.78 | 98.41 | 98.87 | 97.30 | 96.59 | 97.92 |
| does not address time | 96.09 | 91.61 | 96.14 | 98.64 | 98.59 | 94.54 | 98.73 |
| not enough detail | 44.66 | 55.56 | 51.48 | 68.59 | 54.23 | 59.39 | 47.57 |
| ignores science | 84.24 | 77.55 | 86.93 | 90.82 | 81.57 | 90.22 | 94.91 |
| other | 97.66 | 98.30 | 99.09 | 99.77 | 98.12 | 94.77 | 97.80 |
| **uncertainty** | | | | | | | |
| uncertainty missing | 49.35 | 63.46 | 63.07 | 80.02 | 60.10 | 65.39 | 50.57 |
| consensus missing | 70.26 | 75.45 | 66.97 | 81.96 | 77.25 | 81.60 | 82.53 |
| contradicting evidence missing | 92.47 | 88.46 | 94.84 | 95.89 | 91.97 | 92.94 | 95.17 |
| other | 98.18 | 97.74 | 99.20 | 99.77 | 99.03 | 96.06 | 99.31 |

Table 13: Pairwise agreement among the 3 raters per issue. In general we observe high agreement among raters in selecting issues for all models, while some issues such as "vague", "biased", "not enough detail", and "uncertainty missing" are more disagreed upon.

| Issue | InstructGPT | | | ChatGPT | PaLM-2 | GPT4 | Falcon |
|---|---|---|---|---|---|---|---|
| | davinci-002 | davinci-003 | turbo | | text-bison | | 180B-Chat |
| **style** | | | | | | | |
| too informal | 0.85 | 0.96 | 0.99 | 1.00 | 0.93 | 0.94 | 0.96 |
| too long | 0.96 | 0.96 | 0.99 | 0.97 | 0.93 | 0.92 | 0.97 |
| too short | 0.70 | 0.75 | 0.99 | 0.99 | 0.90 | 0.99 | 0.96 |
| inconsistent | 0.81 | 0.96 | 0.99 | 1.00 | 0.96 | 0.99 | 0.98 |
| repetitive | 0.67 | 0.94 | 1.00 | 0.98 | 0.96 | 0.96 | 0.99 |
| other | 0.89 | 0.96 | 0.99 | 1.00 | 0.98 | 0.97 | 0.99 |
| **clarity** | | | | | | | |
| sentences too long | 0.93 | 0.96 | 1.00 | 0.99 | 0.93 | 0.88 | 0.95 |
| too technical | 0.88 | 0.96 | 0.99 | 0.98 | 0.96 | 0.90 | 0.98 |
| hard math | 0.94 | 0.98 | 0.95 | 0.99 | 0.97 | 0.94 | 1.00 |
| other | 0.72 | 0.96 | 0.99 | 1.00 | 0.96 | 0.98 | 0.97 |
| **correctness** | | | | | | | |
| incomplete sentence | 0.89 | 0.94 | 0.99 | 1.00 | 1.00 | 1.00 | 1.00 |
| incorrect spelling | 0.97 | 1.00 | 1.00 | 1.00 | 0.99 | 1.00 | 1.00 |
| incorrect punctuation | 0.92 | 0.98 | 0.98 | 1.00 | 1.00 | 0.99 | 0.98 |
| incorrect grammar | 0.79 | 0.99 | 0.99 | 1.00 | 1.00 | 1.00 | 0.97 |
| other | 0.88 | 0.96 | 1.00 | 1.00 | 0.99 | 1.00 | 0.97 |
| **tone** | | | | | | | |
| biased | 0.21 | 0.18 | 0.34 | −0.03 | 0.14 | 0.18 | 0.28 |
| persuasive | 0.91 | 0.74 | 0.82 | 0.73 | 0.62 | 0.70 | 0.84 |
| negative | 0.95 | 0.93 | 0.94 | 0.96 | 0.82 | 0.90 | 0.93 |
| other | 0.98 | 0.97 | 0.99 | 1.00 | 0.99 | 0.92 | 0.99 |
| **accuracy** | | | | | | | |
| incorrect | 0.40 | 0.79 | 0.82 | 0.91 | 0.67 | 0.85 | 0.85 |
| science out of context | 0.64 | 0.78 | 0.77 | 0.90 | 0.78 | 0.84 | 0.90 |
| self contradictory | 0.91 | 0.97 | 1.00 | 1.00 | 0.97 | 0.98 | 0.99 |
| anecdotal | 0.57 | 0.94 | 0.85 | 0.28 | 0.75 | 0.89 | 0.89 |
| wrong use of terms | 0.93 | 0.95 | 0.97 | 0.99 | 0.94 | 0.98 | 0.99 |
| other | 0.86 | 0.88 | 0.92 | 0.93 | 0.91 | 0.78 | 0.97 |
| **specificity** | | | | | | | |
| irrelevant info | 0.51 | 0.81 | 0.85 | 0.80 | 0.70 | 0.67 | 0.87 |
| vague | −0.02 | 0.08 | 0.21 | 0.18 | 0.05 | 0.14 | 0.21 |
| other | 0.94 | 0.88 | 0.95 | 0.98 | 0.93 | 0.94 | 0.94 |
| **completeness** | | | | | | | |
| does not address main parts | 0.23 | 0.38 | 0.61 | 0.65 | 0.51 | 0.68 | 0.70 |
| does not address region | 0.87 | 0.90 | 0.97 | 0.98 | 0.95 | 0.93 | 0.95 |
| does not address time | 0.92 | 0.83 | 0.92 | 0.97 | 0.97 | 0.89 | 0.97 |
| not enough detail | −0.11 | 0.11 | 0.03 | 0.38 | 0.09 | 0.19 | −0.05 |
| ignores science | 0.68 | 0.55 | 0.73 | 0.82 | 0.63 | 0.80 | 0.90 |
| other | 0.95 | 0.96 | 0.98 | 1.00 | 0.96 | 0.90 | 0.95 |
| **uncertainty** | | | | | | | |
| uncertainty missing | −0.01 | 0.27 | 0.26 | 0.60 | 0.20 | 0.31 | 0.01 |
| consensus missing | 0.41 | 0.51 | 0.33 | 0.64 | 0.53 | 0.64 | 0.65 |
| contradicting evidence missing | 0.85 | 0.77 | 0.90 | 0.92 | 0.84 | 0.86 | 0.91 |
| other | 0.96 | 0.96 | 0.98 | 1.00 | 0.98 | 0.92 | 0.99 |

Table 14: Krippendorff's alpha for agreement on issue selection. The results are consistent with patterns observed in pairwise agreement.

| Issue | InstructGPT | | | ChatGPT | PaLM-2 | GPT4 | Falcon |
|---|---|---|---|---|---|---|---|
| | davinci-002 | davinci-003 | turbo | | text-bison | | 180B-Chat |
| **tone** | | | | | | | |
| biased | 0.84 | 0.60 | 0.55 | 0.58 | 0.34 | 0.35 | 0.43 |
| **accuracy** | | | | | | | |
| incorrect | 0.01 | 0.35 | 0.10 | 0.45 | 0.21 | 0.44 | 0.32 |
| **specificity** | | | | | | | |
| vague | 0.37 | 0.57 | 0.83 | 0.76 | 0.66 | 0.65 | 0.65 |
| **completeness** | | | | | | | |
| does not address main parts | 0.16 | 0.31 | 0.49 | 0.59 | 0.42 | 0.64 | 0.51 |
| not enough detail | 0.30 | 0.62 | 0.77 | 0.87 | 0.63 | 0.72 | 0.71 |
| **uncertainty** | | | | | | | |
| uncertainty missing | 0.55 | 0.81 | 0.77 | 0.91 | 0.83 | 0.80 | 0.72 |
| consensus missing | 0.26 | 0.42 | 0.25 | 0.61 | 0.45 | 0.56 | 0.54 |

Table 15: Krippendorff's alpha for agreement on issue selection, but computed only for low ratings.

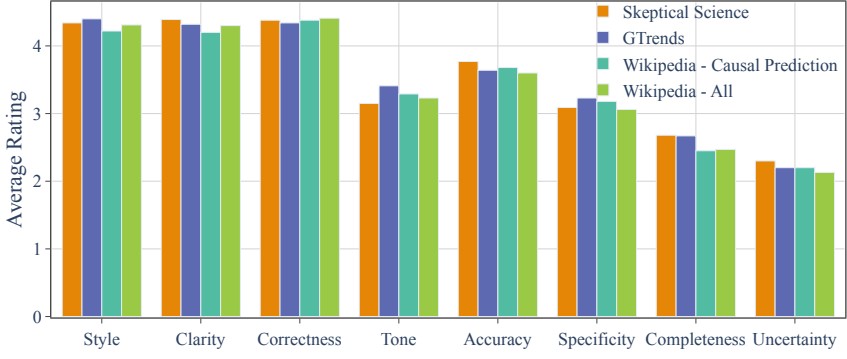

Figure 7: Average rating along all the dimensions per question source and type. In general, there is not a significant difference among the ratings based on the question source or causality.

dimensions. The exception to this rule are answers from *InstructGTP (davinci-002)* which are often shorter and thus quicker to read.

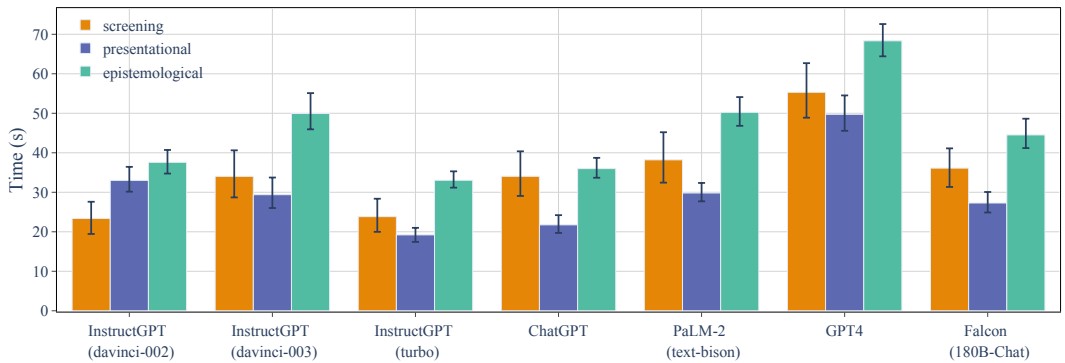

Figure 8: Average time per example for the screening, presentational, and epistemological assessment.

Figure 9 shows that *tone* seems to be harder to assess among the presentational dimensions whereas *accuracy* is quicker among the epistemological dimensions. Otherwise, each dimension takes a similar amount of time.

Larger differences are revealed when we analyze how the rating itself affects the rating times. As expected, Figure 10 shows that high ratings are quicker than lower ones. Keep in mind that for disagreeing ratings (less than 3) we also require the raters to point out specific issues which may add to the length of the interaction. Nevertheless, the trend is also clear among the better (3-5) ratings as well as *between* 1 and 2. For the epistemological dimensions the raters can also select *I don't know*, which takes slightly longer than choosing the middle rating of 3.

## K    QUANTITY OF AI ASSISTANCE

We expect to find a correlation between answer quality and rating, as well as an inverse correlation between answer quality and quantity of AI assistance in each dimension. This is supported by the data, see Figure 11.

## L    LLM RATER

We investigate the possibility of using an LLM to perform the rating task on our evaluation framework. We use `GPT-4` and prompt it using the same language as presented to the human raters (Table 16). We sample 3 responses (temperature 0.6) from `GPT-4` for each question to replicate the setup we have with human raters. We observe the following from the results in Table 17. First, `GPT-4` rater also benefits from assistance. Consistent with findings with human raters, `GPT-4` rater rates answers lower when assistance is provided. This makes sense because (1) the assistance provides an additional chain-of-thought like input and (2) the assistance is generated using additional documents which potentially provide the model with additional information. When the issues in the answers are more severe, `GPT-4` rater agrees with human raters on all dimensions, as evident in the ratings for `InstructGPT` (text-davinci-002). However, `GPT-4` rater disagrees with human raters on the relative ranking of answers from different models of similar quality. Notably, the `GPT-4` rater is more generous towards OpenAI models than humans are.

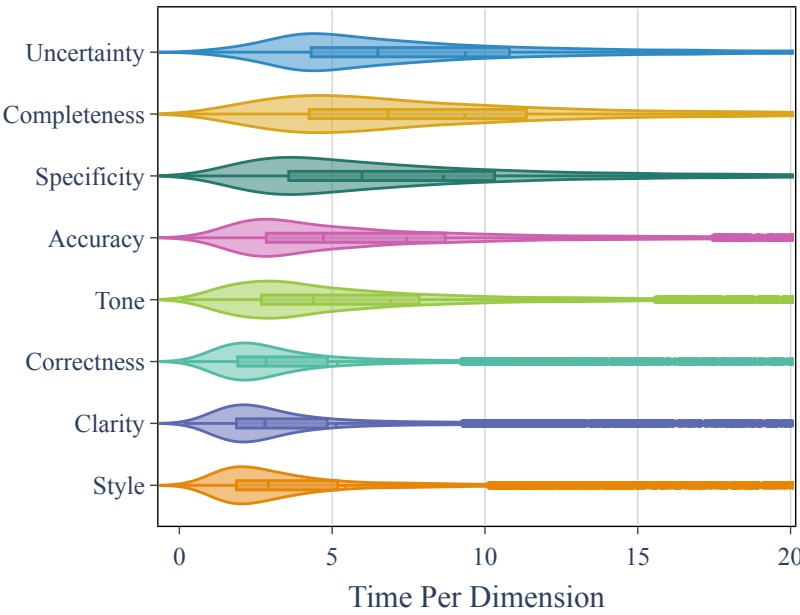

Figure 9: Distribution of rating times for presentational (style, clarity, correctness, tone) and epistemological (specificity, uncertainty, completeness, accuracy) dimensions. For ease of presentation, this figure ignores a small number of timings that took longer than $60s$.

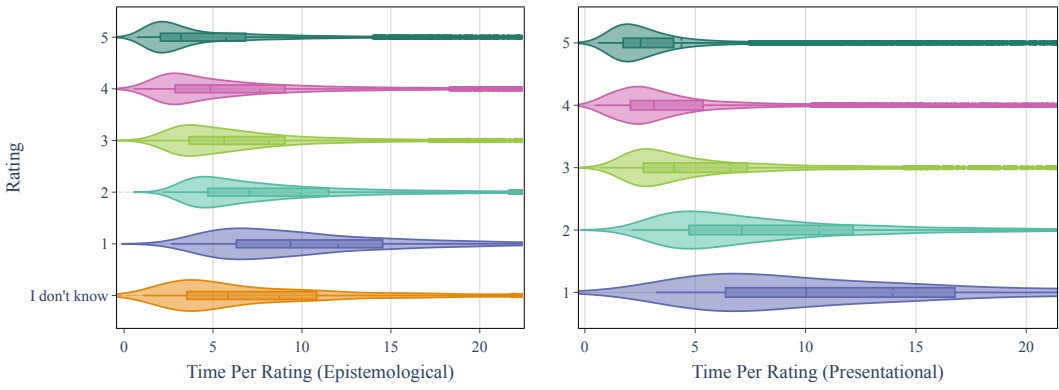

Figure 10: Average time per example for the epistemological and presentational assessment.

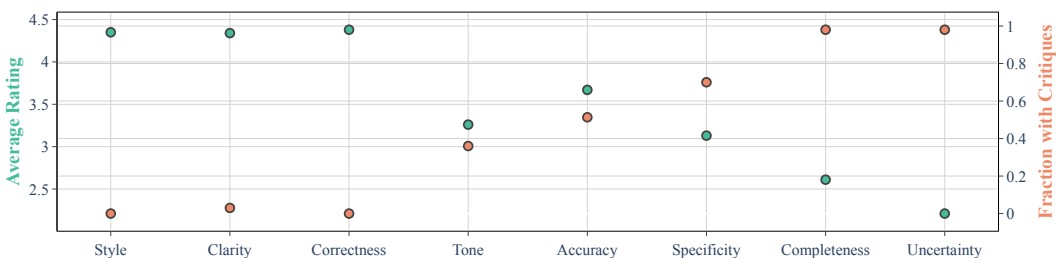

Figure 11: The ratings are lower for dimensions where a higher fraction of the questions has AI Assistance.

| Function | Prompt |
|---|---|
| System | You are an expert on climate communication. Given a question, and an answer to it, and maybe a succinct critique of the answer. |
| Rating | To what extent do you agree with the following statement on the 5-point likert scale (1=disagree completely, 2=disagree, 3=neither, 4=agree, 5=agree completely)?
Question: [*question*]
Answer: [*answer*]
Critique: [*critique*]
Statement: [*statement for different dimensions (below)*]

Answer in the format "Rating: X Problem: Y Explanation: Z" |
| Style Statement | The information is presented well (for a general audience).
If you disagree, what is the problem with the answer? Choose one of the following: too informal/colloquial, answer too long, answer too short, inconsistent language/style/terminology, repetitive, other.
If you choose other, please explain your rating. |
| Clarity Statement | The answer is clear and easy to understand.
If you disagree, what is the problem with the answer? Choose one of the following: sentences too long, language too technical, numbers/formulae hard to understand, other.
If you choose other, please explain your rating. |
| Correctness Statement | The language in the answer does not contain mistakes.
If you disagree, what is the problem with the answer? Choose one of the following: sentence is incomplete, spelling mistakes, punctuation mistakes, grammatical errors, other.
If you choose other, please explain your rating. |
| Tone Statement | The tone of the answer is neutral and unbiased.
If you disagree, what is the problem with the answer? Choose one of the following: the answer is biased, tries to convince me of an opinion/belief, the tone is too negative, other.
If you choose other, please explain your rating. |
| Accuracy Statement | The answer is accurate.
If you disagree, what is the problem with the answer? Choose one of the following: incorrect, takes scientific findings out of context, self-contradictory, anecdotal, wrong use of key terms/scientific terminology, other.
If you choose other, please explain your rating. |
| Specificity Statement | The answer addresses only what the question asks for, without adding irrelevant information.
If you disagree, what is the problem with the answer? Choose one of the following: includes irrelevant parts, too vague/unspecific, other.
If you choose other, please explain your rating. |
| Completeness Statement | The answer addresses everything the question asks for.
If you disagree, what is the problem with the answer? Choose one of the following: misses important parts of the answer, does not address the region the question asks about, does not address time or time range the question asks about, does not give enough detail (e.g., numbers, statistics, details), ignores relevant scientific knowledge, other.
If you choose other, please explain your rating. |
| Uncertainty Statement | The answer appropriately conveys the uncertainty involved.
If you disagree, what is the problem with the answer? Choose one of the following: degree of (un)certainty not given when it should be, agreement in the scientific community not given when important, contradicting evidence (if existing) not mentioned, other.
If you choose other, please explain your rating. |

Table 16: Prompts used to generate ratings.

| System | style | clarity | correctness | tone | accuracy | specificity | completeness | uncertainty |
|---|---|---|---|---|---|---|---|---|
| GPT4 | 4.71 | 4.89 | 5.00 | 3.88 | 4.13 | 3.66 | 2.97 | 2.05 |
| ChatGPT | 4.75 | 4.91 | 4.99 | 3.91 | 4.18 | 3.68 | 2.72 | 2.00 |
| InstructGPT (davinci-003) | 4.39 | 4.68 | 4.63 | 4.05 | 3.49 | 3.29 | 2.44 | 1.91 |
| InstructGPT (davinci-002) | 2.88 | 3.25 | 3.54 | 3.11 | 2.32 | 2.27 | 1.89 | 1.74 |
| InstructGPT (turbo) | 4.62 | 4.82 | 4.89 | 3.80 | 3.76 | 3.30 | 2.46 | 1.94 |
| PaLM-2 (text-bison) | 4.40 | 4.72 | 4.75 | 3.42 | 3.38 | 3.03 | 2.31 | 1.92 |
| Falcon (180B-Chat) | 4.66 | 4.85 | 4.91 | 3.83 | 4.03 | 3.49 | 2.71 | 2.00 |
| GPT4, no assistance | 4.70 | 4.89 | 5.00 | 4.77 | 4.95 | 4.59 | 4.59 | 2.63 |

Table 17: Results from the LLM Rater.

| Age bracket | % |
|---|---|
| [15, 25) | 43.75 |
| [25, 35) | 34.38 |
| [35, 45) | 12.50 |
| [45, 55) | 6.25 |
| [55, 65) | 3.12 |

(a) Distribution of age of our raters.

| Sex | % |
|---|---|
| Female | 56.25 |
| Male | 43.75 |

(b) Distribution of sex of our raters.

| Country of residence | % |
|---|---|
| United Kingdom | 25.00 |
| South Africa | 12.50 |
| Portugal | 12.50 |
| United States | 9.38 |
| Greece | 6.25 |
| New Zealand | 6.25 |
| Netherlands | 6.25 |
| Poland | 6.25 |
| Canada | 3.12 |
| Germany | 3.12 |
| Czech Republic | 3.12 |
| Hungary | 3.12 |
| Italy | 3.12 |

| Ethnicity | % |
|---|---|
| White | 68.75 |
| Black | 12.50 |
| Asian | 12.50 |
| Mixed | 3.12 |
| Other | 3.12 |

(c) Distribution of simplified ethnicities of our raters.

(d) Distribution of countries of residence of our raters.

Table 18: Demographic information of our raters.

# M    RATING FRAMEWORK DETAILS

## M.1    RATER DEMOGRAPHICS

We are working with a group of 32 raters. The raters are all fluent in English and all have at least an undergraduate degree in a climate-related field of study. This includes environmental disciplines (e.g. environmental science, earth science, atmospheric physics, ecology, environmental policy, climate economics), and also other disciplines (including the behavioral and social sciences) as long as their academic work (coursework, project work, or otherwise) involves work on climate or environmental studies. The remaining demographics can be seen in Table 18.

| Presentational Dimensions | Statement and possible issues |
|---|---|
| style | The information is presented well (for a general audience). |
|   too informal | ☐ too informal/colloquial |
|   too long | ☐ answer too long |
|   too short | ☐ answer too short |
|   inconsistent | ☐ inconsistent language/style/terminology |
|   repetitive | ☐ repetitive |
|   other | ☐ other |
| clarity | The answer is clear and easy to understand. |
|   sentences too long | ☐ sentences too long |
|   too technical | ☐ language too technical |
|   hard math | ☐ numbers/formulae hard to understand |
|   other | ☐ other |
| correctness | The language in the answer does not contain mistakes. |
|   incomplete sentence | ☐ sentence is incomplete |
|   incorrect spelling | ☐ spelling mistakes |
|   punctuation mistakes | ☐ punctuation mistakes |
|   incorrect grammar | ☐ grammatical errors |
|   other | ☐ other |
| tone | The tone of the answer is neutral and unbiased. |
|   biased | ☐ the answer is biased |
|   persuasive | ☐ tries to convince me of an opinion/belief |
|   negative | ☐ the tone is too negative |
|   other | ☐ other |

| Epistemological Dimensions | |
|---|---|
| accuracy | The answer is accurate. |
|   incorrect | ☐ incorrect |
|   science out of context | ☐ takes scientific findings out of context |
|   self contradictory | ☐ self-contradictory |
|   wrong use of terms | ☐ wrong use of key terms/scientific terminology |
|   other | ☐ other |
| specificity | The answer addresses only what the question asks for, without adding irrelevant information. |
|   irrelevant info | ☐ includes irrelevant parts |
|   vague | ☐ too vague/unspecific |
|   other | ☐ other |
| completeness | The answer addresses everything the question asks for. |
|   does not address main parts | ☐ misses important parts of the answer |
|   does not address region | ☐ does not address the region the question asks about |
|   does not address time | ☐ does not address time or time range the question asks about |
|   not enough detail | ☐ does not give enough detail (e.g. numbers, statistics, details) |
|   ignores science | ☐ ignores relevant scientific knowledge |
|   other | ☐ other |
| uncertainty | The answer appropriately conveys the uncertainty involved. |
|   uncertainty missing | ☐ degree of (un)certainty not given when it should be |
|   consensus missing | ☐ agreement in the scientific community not given when important |
|   contradicting evidence missing | ☐ contradicting evidence (if existing) not mentioned |
|   other | ☐ other |

Table 19: (on the right) Statements as presented to the raters. We query each dimension separately in the interface (Figure 15) and ask *"To what extent do you agree with the statement below?"* We also require the raters to identify particular issues for the given list if they disagree with a statement. On the left side we list the dimensions the statements belong in and a shorthand for the issue names used in tables throughout this work.

## M.2 RATING STATEMENTS

For presentational and epistemological accuracy we evaluate 4 dimensions each. Given a question-answer pair the raters are asked to what degree they agree with one of the statements in Table 19.[17] The raters select agreement on a 5-point scale from *completely disagree* to *completely agree*. For the two lowest choices we ask for additional details which can be selected from a list of possible issues, including *other* which allows free-text input. See Appendix M.4 for screenshots of the rating interface.

## M.3 TUTORIAL AND ADMISSION TEST

We devise a special introduction session for new participants that contains a tutorial followed by an admission test. The purpose of the session is twofold: (1) The introduction session is designed to familiarize the raters with the interface and the task. (2) Based on the session's outcome we select raters into the rating pool.

**Tutorial** In the tutorial (see Figure 12) we present 4 examples of increasing difficulty in the rating interface and only ask for one dimension each. Each example exhibits a particular main issue and we expect raters to identify this issue correctly. A hint is given if the rater selects a wrong answer that does not identify the issue and they can only proceed to the next item if an acceptable answer that does identify the issue is given. Regarding other issues than the main issue, one might disagree on some of these issues and we allow several possible selections. Note that to identify the main issue, a low rating (disagree completely or disagree) must be selected. Once a valid response is selected we show positive feedback and explain why the outcome is the desired one. We don't collect any data during the tutorial part.

**Admission Test** To test the raters' ability and attention to detail we select three realistic examples that exhibit at least one major flaw. We use the full template and ask about all statements in Table 19. We record the responses and assign or deduct points for every detected, undetected, and over-detected issue. The point scheme was decided among the authors after carefully considering possible disagreements or subjective interpretations.

Based on the performance of an early group of raters with known performance on the task we decide on a threshold and admit raters above that score to the pool. We believe that the tutorial and admission test were effective in ensuring that raters were both familiar with the interface as well as the type of assessment we expect from them, which requires close reading of question and answer, basic knowledge of climate change, and an understanding of the tasks dimensions and issues that allows them to rate dimensions and select specific issues reliably.

## M.4 TEMPLATE SCREENSHOTS

We provide screenshots of our rating implementation in Figures 13 to 15.

# N EXAMPLES

## N.1 ANSWERS

In this section, we present examples showing good versus bad answers for each dimension evaluated in Tables 20 to 27.

## N.2 AI ASSISTANCE

In this section, we present examples showing AI assistance that influence human ratings in Tables 28 to 32.

---

[17]Please note that when we use the shorthand *correctness* in our results, this only refers to correctness *of the language*, i.e. presentational correctness. The corresponding epistemological dimension is *accuracy*, i.e. correctness *of the answer*.

**Question:** Is SO2 a tool to mitigate global warming?

**Answer:** SO2 cannot be used as a tool to mitigate global warming. It is a pollutant emitted from burning fossil fuels, and has negative effects on human health and the environment. However, in some cases, it has been suggested that injecting SO2 into the atmosphere could be used as a geoengineering technique to reflect sunlight back into space and temporarily cool the planet. This approach, known as stratospheric sulfate aerosol injection, is highly controversial and comes with significant risks and uncertainties. It is considered by experts as a potential last resort option, if all other climate mitigation efforts fail.

To what extend do you agree with the statement below?

The answer is accurate.

○ disagree completely    ○    ○    ○    neither    ○ agree completely    ○ I don't know

Please provide more detail (multiple options possible):

☑ incorrect

☐ takes scientific findings out of context

☑ self-contradictory

☐ anecdotal

☐ wrong use of key terms/scientific terminology

☐ other

Good job! The answer says that SO2 *cannot* be used to mitigate global warming, while it is also considered a *last resort option*. This cannot both be true, so the answer is self-contradictory and must be at least partially incorrect.

◀ BACK          ✓ COMPLETE & START RATING

Figure 12: Screenshot of the last of 4 tutorial questions with the correct answer selected.

## Introduction

You are being invited to participate in a research study. We are a research team interested in using AI to provide better answers to questions about climate change, according to communication science principles. We plan to publish the data obtained with this questionnaire together with a research paper on the subject.

### What will I have to do?

You are shown a question and an answer to the question. Please read both the question and the answer carefully. Your task is to help us determine how good the answer is, and whether it is supported by references. To do that, we will ask you to rate a series of statements (see below).

For some statements, AI assistance will be provided. Note that while the assistance is meant to be helpful, it may not be helpful or it may even be incorrect. We will also ask you about the helpfulness of the assistance.

The task should be self-contained. It is OK to quickly look something up, but beyond that please select "I don't know" if you can't decide based on the provided information, rather than researching additional resources like search engines or other sources.

### What will I be asked?

You will be asked to assess the answer in several dimensions, rating the following statements (in bold):

**Presentation**

- **(Presentation) The information is presented well for a general audience.** In particular, the answer is not too long or too short, there is no repetition in the text, and the answer is not too informal or too technical.
- **(Understandability) The answer is clear and easy to understand.** Sentences are not too long / verbose or too short. If there are numbers and formulae, they are easy to understand.
- **(Correctness) The language in the answer does not contain mistakes.** In particular, there are no grammatical, spelling, or punctuation errors.
- **(Tone) The tone of the answer is neutral and unbiased.** In particular, the tone is not negative and the answer does not try to convince the reader of an opinion or belief.

**Epistemology**

- **(Accuracy) The answer is accurate.** Whether the answer is correct. More specifically, it does not take scientific findings out of context, does not contradict itself, does not rely on anecdotal evidence, and does not misuse key terms or scientific terminology.
- **(Specificity) The answer addresses only what the question asks for, without adding irrelevant information.** There is no irrelevant statement with respect to the question in the answer, and there is no vague or generic statement in the answer.
- **(Completeness) The answer addresses everything the question asks for.** In particular, it does not miss any part of the question and provides enough necessary details, e.g., numbers, statistics, and details. If the question asks for a specific time range or region, the answer correctly provides that information.
- **(Uncertainty) The answer appropriately conveys the uncertainty involved.** If the scientific community is uncertain about something mentioned in the answer, that uncertainty is conveyed appropriately in the answer. Note that it may be appropriate not to mention uncertainty at all.

**Attribution**

In the final section, you will be asked to assess whether the answer is supported by passages from Wikipedia.

**Providing detail**

If you give a low rating (disagree or disagree completely), we will ask you to provide additional details. You can either select reasons from a list, or provide additional information in a text box. When offering free text, please be specific about why you gave a particular answer a low rating. This will help us add more options in the future.

Thank you for helping us provide better answers to questions about climate change! If you have any feedback or comments on how to improve this questionnaire, please send us a message on the platform.

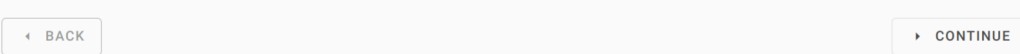

Figure 13: Screenshot of the instructions to the raters, provided at the beginning of the first rating session.

**Instructions:** You are shown a question about climate change and an answer to the question. Please read both question and answer carefully.

For some questions, AI assistance will be provided. Note that while the assistance is meant to be helpful, it may not be helpful or even incorrect. We will also ask you about the helpfulness of the assistance.

The task should be self-contained. It is OK to quickly look something up, but beyond that please select 'I don't know' if you can't decide based on the provided information, rather than exploring additional resources like search engines or other sources.

**Question:** If climate change in the jurassic time was good for the dinosaurs, could similarly slow global warming be good for us as well?

**Answer:** While it's true that the Jurassic period was a time of high carbon dioxide levels and warm temperatures that were beneficial for dinosaurs, it's important to remember that the Earth's ecosystems and species have significantly evolved since then. The rate of current global warming is much faster than most past natural climate changes. This rapid change is what's most concerning for humans and current biodiversity, as it doesn't provide enough time for species to adapt. Furthermore, human societies and infrastructure are designed for the relatively stable climate we've had for the past few thousand years. Rapid and significant changes in that climate present serious risks to our societies.

First we need to **make sure** you understand the question and the answer well enough to evaluate them further. (It is OK if you don't understand some of the more technical language or mathematics, you are not expected to be an expert on climate science. This is not a test of your knowledge, and we are interested in your honest assessment of the answer.)

**I understand the question.**   ● Yes   ○ No

**I understand the answer.**   ● Yes   ○ No

**The answer addresses at least some part of the question.**   ● Yes   ○ No

Figure 14: Screenshot of screening template which ensures the raters are able to judge the example. If any of the questions is answered *No* we skip the example and proceed with the next one.

| Question | In what ways can peri-urban ecosystems help reduce the urban heat island effect? | |
|---|---|---|
| | **Model A** | **Model B** |
| Answer | Peri-urban ecosystems can significantly help reduce the urban heat island effect in several ways. Firstly, they act as natural coolants by providing shade and releasing moisture into the atmosphere through a process known as evapotranspiration, which can lower surrounding temperatures. Secondly, these ecosystems, which include forests, grasslands, and wetlands, can absorb and store heat, reducing the overall temperature in urban areas. Lastly, they can also help mitigate the heat island effect by reducing the need for energy-intensive cooling systems, as well-vegetated areas are naturally cooler than concrete or asphalt surfaces. | Peri-urban ecosystems can help reduce the urban heat island effect by providing shade, evaporative cooling, and habitat for cooling breezes. Trees and other vegetation in peri-urban areas can help to block the sun's rays, reducing the amount of heat that is absorbed by the ground. The evaporation of water from plants and soil can also help to cool the air, and vegetation can create shade that helps to block the sun's heat. In addition, vegetation can provide habitat for animals that help to create cooling breezes. All of these factors can help to reduce the urban heat island effect, making cities more livable and sustainable. |
| Assistance | | The answer contains repetition, specifically in the explanation of how vegetation provides shade and blocks the sun's heat. |
| Issues | | repetitive |
| Rating | 4.6 | 2.0 |

Table 20: Example for the **style** dimension. The Model B answer repeats the explanation on how vegetation provides shade and blocks the sun's heat.

Figure 15: Screenshot of the rating interface. Here we are rating the *completeness* dimension. Since the rater selected a low rating, the interface asks for specific details. For ratings of *3 = neither* and above, the blue box is hidden. In this example we also show the AI Assistance which is not always available, which is rated as not helpful.

| Question | Was the name 'global warming' changed to 'climate change'? | |
|---|---|---|
| | **Model A** | **Model B** |
| Answer | Yes | Yes, the term "global warming" was changed to "climate change" in order to better reflect the broader range of impacts associated with rising greenhouse gas emissions. While global warming refers specifically to the increase in Earth's average surface temperature, climate change encompasses a wider array of changes, including shifts in precipitation patterns, sea level rise, and the frequency and intensity of extreme weather events. The term "climate change" is now widely used to capture the full scope of these impacts. |
| Assistance | The answer lacks sufficient detail to fully address the complexity of the question. | |
| Issues | too_short, vague | |
| Rating | 2.6 | 4.6 |

Table 21: Example for the **clarity** dimension. The Model A answer is too short. It lacks details and context to fully address the question.

| Question | Which areas in Nigeria are at risk of rising sea levels? | |
|---|---|---|
| | **Model A** | **Model B** |
| Answer | The coastal areas of Nigeria are at risk of rising sea levels due to climate change. These areas include Lagos, Ogun, and Delta states. Rising sea levels could cause flooding, erosion, and salinization of freshwater sources. This could have a devastating impact on the livelihoods of the people who live in these areas. | I do not have access to the latest information on the specific areas in nigeria that are at risk of rising sea levels. however, it is important to note that nigeria, like many other countries, is vulnerable to the impacts of climate change, including sea level rise. coastal cities and communities are particularly at risk, and it is crucial for nigeria to take action to mitigate and adapt to the effects of climate change. |
| Assistance | | The answer contains a grammatical error - "nigeria" should be capitalized as "Nigeria". |
| Issues | | incorrect_grammar, incorrect_spelling |
| Rating | 4.6 | 2.3 |

Table 22: Example for the **correctness** dimension. The Model B answer contains a grammatical error where the first letter of a location name should be capitalized.

| Question | Do volcanos emit more CO2 than humans? | |
|---|---|---|
| | **Model A** | **Model B** |
| Answer | No, volcanoes do not emit more CO2 than humans. According to the U.S. Geological Survey, human activities, particularly the burning of fossil fuels for energy, emit more than 60 times the amount of carbon dioxide than volcanoes each year. While volcanoes do contribute to CO2 emissions, their contribution is relatively small compared to human activities. It's important to note that human-induced climate change is primarily driven by the increase in greenhouse gases from human activities, not natural processes like volcanic eruptions. | No, volcanoes do not emit more CO2 than humans. While volcanic eruptions do release significant amounts of CO2 into the atmosphere, the total annual emissions from volcanoes are estimated to be about 0.3 to 0.4 billion metric tons. In contrast, human activities, such as burning fossil fuels and deforestation, emit over 35 billion metric tons of CO2 annually, which is about 100 times more than volcanic emissions. |
| Assistance | | |
| Issues | biased, persuasive | |
| Rating | 2.0 | 4.3 |

Table 23: Example for the **tone** dimension. Instead of answering with only facts, the Model A answer assigns blame on human activities as the cause of climate change which was not part of the question.

| Question | How does burning fossil fuels mitigate climate change | |
|---|---|---|
| | **Model A** | **Model B** |
| Answer | Burning fossil fuels does not mitigate climate change. In fact, it is one of the primary causes of climate change. When fossil fuels are burned, they release carbon dioxide and other greenhouse gases into the atmosphere, which trap heat and contribute to global warming. To mitigate climate change, we need to transition to cleaner, renewable sources of energy and reduce our overall energy consumption. | Burning fossil fuels releases greenhouse gases into the atmosphere, which trap heat and cause the planet to warm. This is the leading cause of climate change. However, burning fossil fuels can also be used to mitigate climate change by generating electricity from renewable sources, such as solar and wind power. This can help to reduce the amount of greenhouse gases emitted into the atmosphere and slow the rate of climate change. |
| AI Assistance | | The critique is that the statement inaccurately suggests that burning fossil fuels can mitigate climate change. As per the provided paragraphs, "the burning of fossil fuels is the main source of greenhouse gas emissions causing global warming and ocean acidification." Furthermore, "the International Energy Agency concluded that no new fossil fuel extraction projects could be opened if the global economy and society wants to avoid the worst impacts of climate change and meet international goals for climate change mitigation." Therefore, burning fossil fuels contributes to climate change rather than mitigating it. |
| Issues | | incorrect, science_out_of_context, self_contradictory |
| Rating | 4.6 | 1.3 |

Table 24: Example for the **accuracy** dimension. The Model B answer contains a factual error which suggests that burning fossil fuels can mitigate climate change.

| Question | What happened to global temperatures by the mid-1970s? | |
|---|---|---|
| | **Model A** | **Model B** |
| Answer | By the mid-1970s, global temperatures began to rise at an accelerated rate. This period marked the end of a slight cooling phase that had occurred from the 1940s to the 1970s, often referred to as the "global cooling" period. The shift in temperature trends was largely due to increased levels of greenhouse gases, particularly carbon dioxide, in the Earth's atmosphere from human activities such as burning fossil fuels. This marked the beginning of a period of rapid global warming that continues today. | By the mid-1970s, global temperatures had risen by about 0.2 degrees Celsius (0.36 degrees Fahrenheit) since the late 19th century. This warming trend was statistically significant, and it was consistent with the predictions of climate models that had been developed in the 1960s and 1970s. |
| Assistance | The statement is incorrect in asserting that there are no vague or generic statements in the answer. The phrase "increased levels of greenhouse gases, particularly carbon dioxide, in the Earth's atmosphere from human activities such as burning fossil fuels" is a generic statement that could benefit from specific data or examples. As per Wikipedia, "The 2011–2020 decade warmed to an average 1.09 °C [0.95–1.20 °C] compared to the pre-industrial baseline (1850–1900). Surface temperatures are rising by about 0.2 °C per decade, with 2020 reaching a temperature of 1.2 °C above the pre-industrial era." This specific information could enhance the answer's clarity and precision. | The answer could be improved by including the specific information that "The 2011–2020 decade warmed to an average 1.09 °C [0.95–1.20 °C] compared to the pre-industrial baseline (1850–1900)" and "Surface temperatures are rising by about 0.2 °C per decade," which provides a more precise context for the warming trend observed by the mid-1970s. |
| Issues | vague | |
| Rating | 2 | 4 |

Table 25: Example for the **specificity** dimension. The Model A answer omits specific numbers on how much the global temperatures have changed which is important for the answer's clarity and precision.

| Question | How much do cows contribute to climate change | |
|---|---|---|
| | **Model A** | **Model B** |
| Answer | Cows significantly contribute to climate change due to the production of methane, a potent greenhouse gas, during their digestion process. According to the United Nations' Food and Agriculture Organization, livestock (including cows) account for about 14.5% of all human-caused greenhouse gas emissions, with cows being the largest contributor. This is equivalent to the exhaust emissions from all the vehicles on the planet. Therefore, the impact of cows on climate change is substantial and cannot be overlooked. | Cows are a major source of methane, a powerful greenhouse gas that contributes to climate change. According to the Environmental Protection Agency (EPA), cattle and other livestock are responsible for about 5.5 percent of all human-caused greenhouse gas emissions. |
| Assistance | While the answer provides a comprehensive overview of cows' contribution to climate change, it fails to mention the potential for mitigation efforts, such as the adoption of best practices that can reduce emissions by up to 30%, as stated by the FAO. Furthermore, it does not address the variability in environmental impacts due to different agricultural practices around the world. | While the answer provides a general overview of the contribution of cows to climate change, it fails to mention the potential mitigation strategies, such as dietary changes and adoption of best practices, which can significantly reduce emissions. As stated in the provided paragraphs, "Mitigation options for reducing methane emission from livestock include a change in diet, that is consuming less meat and dairy" and "FAO estimates that the adoption of already available best practices can reduce emissions by up to 30%". |
| Issues | | not_enough_detail, does_not_address_main_parts, ignores_science |
| Rating | 4.6 | 2 |

Table 26: Example for the **completeness** dimension. The Model B answer lacks context and details around the answer. The assistance on missing mitigation efforts is not helpful because it is not necessary to make the answer complete. The raters are able to recognize this and give Model A answer a correct high rating.

| Question | Did Hansen predict that the West Side highway would be underwater in 20 years? | |
|---|---|---|
| | **Model A** | **Model B** |
| Answer | James Hansen, a prominent climate scientist, did make a statement in a 1988 interview suggesting that the West Side Highway in New York City could be underwater in 40 years. However, this was not a formal prediction based on a specific scientific study, but rather a speculative comment made to illustrate potential impacts of climate change. It's important to note that while sea levels are indeed rising due to climate change, the exact rate and impact can vary greatly depending on a range of factors, including greenhouse gas emissions and ice melt rates. | No, Hansen did not predict that the West Side highway would be underwater in 20 years. He did, however, predict that sea levels would rise by about 2 feet by 2100, which would put the West Side highway at risk of flooding. |
| Assistance | The statement overlooks the importance of communicating uncertainty in climate science. As per the answer, Hansen's comment was speculative and not based on a specific study, thus it's crucial to mention this uncertainty to avoid misinterpretation of the potential impacts of climate change. | The statement fails to acknowledge that the answer does not explicitly convey the inherent uncertainty in climate predictions, such as Hansen's prediction about sea level rise. According to Wikipedia, "Uncertainty is a fundamental characteristic of weather, seasonal climate, and hydrological prediction, and no forecast is complete without a description of its uncertainty." |
| Issues | | uncertainty_missing |
| Rating | 4 | 2.3 |

Table 27: Example for the **uncertainty** dimension. The Model B answer states precise predictions without acknowledging the uncertainty in climate science as there are many factors that can influence a prediction.

| Question | Is IPCC consensus phoney? |
|---|---|
| Answer | No, the consensus of the Intergovernmental Panel on Climate Change (IPCC) is not phoney. The IPCC is a reputable international body that assesses the science related to climate change. It was established by the United Nations Environment Programme (UNEP) and the World Meteorological Organization (WMO) in 1988 to provide policymakers with regular scientific assessments on climate change, its implications and potential future risks, as well as to put forward adaptation and mitigation strategies. The consensus reached by the IPCC is based on rigorous scientific research and review by experts from all over the world, making it a reliable and authoritative source of information on climate change. |
| Assistance | The tone of the answer, while factual, may be perceived as biased due to the strong defense of the IPCC's credibility and authority. |
| Rating with Assistance | 2.3 [biased, persuasive] |
| Rating without Assistance | 4.6 |

Table 28: Example for the **tone** dimension. Raters identify the biased tone only with assistance.

| Question | How do single-family houses contribute to higher toxic chemical emissions compared to multifamily buildings? |
|---|---|
| **Answer** | Single-family houses contribute to higher toxic chemical emissions compared to multifamily buildings primarily due to their larger size and individualized systems. Each single-family home typically has its own heating, cooling, and water systems, all of which consume energy and produce emissions. Additionally, these houses often have larger lawns requiring maintenance equipment like lawnmowers and leaf blowers, which also emit harmful chemicals. Lastly, single-family homes, due to their larger size, require more materials to build and maintain, leading to higher emissions from the production and transportation of these materials. |
| **Assistance** | While the answer provides a plausible explanation, it lacks empirical evidence or scientific findings to support the claim that single-family houses contribute to higher toxic chemical emissions compared to multifamily buildings. |
| **Rating with Assistance** 
 **Rating without Assistance** | 1.6 [anecdotal, incorrect, science_out_of_context] 
 4 |

Table 29: Example for the **accuracy** dimension. Raters identify the anecdotal nature of the answer only with assistance.

| Question | Are climate change solutions too expensive? |
|---|---|
| **Answer** | While it's true that implementing solutions to combat climate change requires significant financial investment, it's important to consider the cost of inaction, which is likely to be much higher. The economic damage caused by extreme weather events, rising sea levels, and other climate-related impacts could be devastating. Moreover, many climate solutions, such as renewable energy technologies, are becoming increasingly cost-competitive. Investing in these solutions can also stimulate economic growth and create jobs, providing a return on investment. Therefore, while the upfront costs may be high, the long-term benefits both economically and environmentally make it a necessary investment. |
| **Assistance** | The statement is incorrect as the answer contains a vague statement. The phrase "the cost of inaction, which is likely to be much higher" is not quantified or supported with specific data or examples. As per the provided Wikipedia paragraphs, "a 2021 study by the reinsurance company Swiss Re estimated global climate change is likely to reduce global economic output by 11-14%, or as much as $23 trillion annually by 2050, compared with output without climate change." This specific information could have been used to substantiate the claim. |
| **Rating with Assistance** 
 **Rating without Assistance** | 2 [vague, irrelevant_info] 
 4.6 |

Table 30: Example for the **specificity** dimension. Raters identify how specific information could make the answer better only with assistance.

| Question | Is the Arctic ice melt a natural cycle? |
|---|---|
| **Answer** | While the Arctic ice does go through natural cycles of melting and refreezing, the accelerated rate of ice melt that we are currently witnessing is not part of a natural cycle. Scientific evidence overwhelmingly points to human-induced climate change as the primary cause. Rising global temperatures, largely due to greenhouse gas emissions from human activities, are causing the Arctic ice to melt at an unprecedented rate. This rapid ice loss is having profound effects on Arctic ecosystems and contributing to rising sea levels globally. |
| **Assistance** | While the answer provides a comprehensive explanation of the Arctic ice melt, it lacks specific details such as the decline rate of 4.7% per decade, the projection of the first ice-free conditions around 2035 under a scenario of continually accelerating greenhouse gas emissions, and the fact that Arctic ice decline between 1979 and 2011 is estimated to have been responsible for as much radiative forcing as a quarter of CO2 emissions the same period. |
| **Rating with Assistance** | 2 [not_enough_detail] |
| **Rating without Assistance** | 5 |

Table 31: Example for the **completeness** dimension. Raters identify how specific information could make the answer better only with assistance.

| Question | will global warming cause an ice age |
|---|---|
| **Answer** | While it may seem counterintuitive, some scientists have suggested that global warming could potentially trigger what's known as a "mini ice age." This theory is based on the idea that increased melting of polar ice could disrupt ocean currents, such as the Gulf Stream, that play a crucial role in regulating global climate. If these currents were to be significantly altered or shut down, it could potentially lead to a dramatic cooling of certain regions, such as Northern Europe. However, this is just one theory and the overall consensus among scientists is that global warming will lead to a general trend of increasing temperatures worldwide. |
| **Assistance** | While the answer acknowledges the theory of a potential "mini ice age" due to global warming, it fails to adequately convey the scientific consensus on the matter. As stated in the provided paragraphs, "Earth system models used in Coupled Model Intercomparison Project indicate that shutdown is only likely after high levels of warming are sustained well after 2100," and "paleoceanographic research suggests that the AMOC may be even more stable than what is predicted by most models." Therefore, the answer could mislead readers into thinking that a "mini ice age" is a more immediate and likely consequence of global warming than what is generally agreed upon by the scientific community. |
| **Rating with Assistance** | 1.6 [uncertainty_missing, consensus_missing, contradicting_evidence_missing] |
| **Rating without Assistance** | 4.6 |

Table 32: Example for the **uncertainty** dimension. Raters identify the unlikelihood of the theory only with assistance.

