# OpenReview forum: "Assessing Large Language Models on Climate Information"
_ICLR.cc/2024/Conference — Submitted to ICLR 2024_

### Official Review · Reviewer_yjrw · 2023-10-30

**Soundness:** 3 good
**Presentation:** 4 excellent
**Contribution:** 3 good
**Rating:** 8
**Confidence:** 4

**Summary:**

The paper proposes an evaluation framework for judging climate information generated by large language models. The evaluation mainly focuses on presentation skills and correctness, measured with four metrics each. Climate related questions are generated from three different sources, including from GPT-4 using Wikipedia articles. State-of-the-art LLMs are used to generate content. Evaluation is done with educated humans, and experiments include assistance with AI for evaluation. Overall, LLMs don't do well in communicating at the right tone, and do not convey uncertainty information properly.

**Strengths:**

- A very relevant problem statement for the AI community given that LLMs are being used to simplify complicated concepts, including climate change.
- A good summary of relevant literature and identification of metrics corresponding to presentation and epistemology
- Well constructed dataset and evaluation mechanism with educated human annotators
- Evaluation of human evaluators is explained clearly
- Using LLMs to assist humans in evaluation is a good addition.

**Weaknesses:**

- The LLM prompt does not state the evaluation criteria on presentation and epistemology. If all the eight metrics were included in the prompt, it is likely the response quality will improve (my hypothesis). With the current prompt, the LLM is "unaware" that it is being evaluated this way. An ablation with and without such a prompt/instruction will also be interesting to see.
- The LLM is not evaluated against a human expert output. It would be good to see an expert answer the same question, and evaluate them on the eight metrics. Currently, the 1-5 score grid is an arbitrary bar that the LLM is being held to without comparing with existing systems.
- A discussion of wait time and cost of running this LLM experiment will be good to see. Especially, comparing these against existing human based systems will be interesting.
- Claims in the sub-section "Resolution and Range" do not have sufficient evidence.
- Concerns about audience context is raised multiple times, but is not addressed in the paper

**Questions:**

1. Why did you use a 3-4 sentence restriction? Instead, it would have been better to ask the evaluators if the response was too long.
2. Is GPT-4 used for all of the AI assistance generated, or do you use respective LLMs to create the assistance paragraph?
3. Do all the LLMs support the input length to supply the paragraphs in the Wikipedia articles as input? It may be that paragraphs in the lower sections get ignored given the input text limit.
4. What happens if answers to the questions are not available in Wikipedia? Are there such questions in your dataset?
5. Do you include questions to which do not have answers to from the latest science?
6. I didn't understand the concept of "previous exposure". Don't I have exposure as soon as I evaluate one question? From that respect, all but the first question have "previous exposure"

---

> ### Author Response · Authors · 2023-11-16
> **Response to Reviewer yjrw**
>
> We thank the reviewer for the encouraging feedback and good suggestions which we are trying to implement.
>
> **Specific weaknesses:**
>
> 1 - Thanks for the suggestion, we are trying to run an experiment where we augment the prompt with a description on the evaluation criteria. We hope to be able to update the paper with these results as soon as possible. Notice, however, that this is a somewhat of a theoretical factor. A generalist model (the object of this study) may be evaluated in many different ways on many different dimensions and topics and cannot be expected to be able to condition at answer generation time on the evaluation dimensions. In fact, some of the dimensions may have conflicting goals (say ‘creative tasks’ vs ‘scientific information’). At the same time, it is not realistic to expect that users will do that kind of ‘prompt tuning’ while asking questions.
>
> 2 - In preliminary experiments we found that evaluating actual experts' outputs is quite complex. This will be one of the main things we will focus on in the next steps. However, we are trying to see if we could get at least some evidence in that direction during the rebuttal period.
>
> 3 - We are not sure what the request is regarding wait times and cost. Is this about the cost of calling the API? Or about generation of AI assistance? For our experiments the API costs are tiny compared to rating expenses.
>
> 4 - We wonder why the reviewer thinks this is not the case. We would welcome more details. In terms of range we, as a first approximation, consider the difference between the best model score and the worst model score. This turns out to be several times the magnitude of the standard deviation, and we interpret this as providing evidence for sufficient range. In terms of resolution, our results show that the model's performance, e.g. for the GPT models, is sufficiently aligned with expectations: newer models are better than older models, with the oldest (InstructGPT-002) performing the worst.
>
> 5 - We are not exactly sure what the reviewer is asking about. For our evaluation, we assume that all 'audience context' is part of the question, i.e. we don't ask our raters to assume a specific context. However we do assess whether the answer is sufficiently specific to the question, e.g. if the question is about sea level rise around Iceland in the year 2100, the answer should address the relevant geographic region and time.
>
> **Specific questions:**
>
> 1 - We add the 3–4 sentence length specification to provide a minimum of control on the maximum length of the answer, thus reducing the variance in answer form. This also helps to reduce the variance in raters' performance because longer answers may be more difficult to rate.
>
> 2 - Only GPT-4 is used to produce the auxiliary information, to avoid confounding factors due to the differing abilities of a specific model in performing them.
>
> 3 - We limit the Wikipedia article evidence to the top 3 passages, so capacity is not an issue. However, LLMs don’t need the Wikipedia articles to answer the question. The Wikipedia article is only used to generate the model assistance for the epistemological dimensions and we always use GPT-4 to generate the assistance, see above.
>
> 4 - If there is no Wikipedia document relevant for the answer (which happens in about 10% of the answers) we simply do not generate AI assistance for the epistemological dimensions and such answers are not used for the AIS evaluation study. In any case, the raters don’t need the Wikipedia article for the presentational and epistemological rating task.
>
> 5 - Novel questions. We don’t have a dedicated set of this kind and we are not aware of questions of this kind in the current dataset. However, we believe this is a good suggestion and we will try to address this point in the next version of the data. This requires careful thought and validation.
>
> 6 - Previous exposure. Previous exposure means that a rater has seen and rated some answers where AI assistance was shown. The three categories work as follows:
>
> 1. *‘With AI Assistance’:* Answers in the evaluated set are accompanied by AI assistance and the raters are shown the assistance.
> 2. *‘Without AI Assistance but previous exposure’:* None of the answers in an evaluated set are associated with AI assistance. The raters assess the answers without being provided any AI assistance. However, in previous studies, the same raters have evaluated answers accompanied by AI assistance. What our results show is that this (previous exposure) seems to have a training effect on the raters and they are able to spot more issues compared to the ‘With AI Assistance’ setting.
> 3. *‘Without AI Assistance’:* A fresh set of raters assesses the evaluated set, for the first time, without ever having rated any answers that were provided with AI assistance. In this setting, (Figure 3a) raters recognize the least issues.
>
> We will make this clearer in the paper.

---

> ### Comment · Reviewer_yjrw · 2023-11-22
> **Thanks for the responses**
>
> The responses make sense to me. Clarifying few aspects I had asked for:
>
> 3. For cost, I was expecting the amount of money and time spent to get a response from an LLM vs the same response from an expert.
>
> 4. I was expecting a statistical test to make conclusions
>
> Overall, enjoyed reading the paper, thanks!

---

> > ### Author Response · Authors · 2023-11-23
> > **Response 2 to Reviewer yjrw**
> >
> > Thank you for the valuable feedback!
> >
> > We uploaded an updated version of the paper, making changes to the paragraph on "Resolution and Range" in Section 3.4. Most importantly, we have clarified what we mean by resolution and range.
> >
> > We are also working on a more substantial change in response to the two points you clarified. In particular, we will provide a table of statistical tests across dimensions and models. In the meantime we believe the paragraph has been significantly improved already.

---

### Official Review · Reviewer_6F8Y · 2023-11-02

**Soundness:** 2 fair
**Presentation:** 3 good
**Contribution:** 1 poor
**Rating:** 3
**Confidence:** 5

**Summary:**

The paper uses LLMs as a Q-A chat bot for disseminating climate information. It lists different criteria and evaluates different LLMs on those criteria.

**Strengths:**

I especially like this application since it relates to an important world problem. The desiderata for good climate communication are a key contribution. The authors present criteria based on presentational and epistemological adequacy. Such a framework can be generalized to other domains for LLM evaluations, so I really appreciated this section.

The paper is well-written and evaluation of LLMs has enough discussion.

**Weaknesses:**

However, a key weakness of the work is that lack of any prompt tuning. The prompt provided is a 2-sentence prompt that does not include any of the known techniques that improve LLMs' performance. I would suggest the authors to include

1) in-context examples that can be random or retrieved based on query similarity. This will help both grounding and help with tone--a factor that most LLMs were weak at. This is probably because LLM does not the expected tone---if some examples are provided, then LLM output may be similar.
2) At a minimum, the presentational and epistemic criteria can also be included in the prompt. For example, if there is a need for a certain style (e.g., text should not informal), that can explicitly be included in the prompt.

Without basic prompt enhancements like these, the results may not an accurate assessment of the models. For instance, I suspect that the results on Tone will increase significantly after these changes.

Evaluating LLM system is a combination of prompt and model. I suggest that the authors include 3-4 prompt variations and compare models to get more robust answers.

**Questions:**

1) What will happen to Tone results if you include in-context examples?
2) Can you show ablations on Figure 2 with the prompts suggested above?

---

> ### Author Response · Authors · 2023-11-16
> **Response to Reviewer 6F8Y**
>
> We thank the reviewer for their comments and suggestions. Here we provide some initial answers. In the remaining rebuttal period, we will be trying to run some of the experiments suggested. Our goal is to integrate the results and discussion in an updated version of the paper as soon as possible.
>
> To the specific point: We agree that prompt tuning, especially specific to each model, is likely to improve results. In this work our priority is mainly around defining the evaluation framework and protocol. For the first results we did not include prompt tuning in the experiments and instead compared several models using an identical prompt. Note that this is likely the most common way for people to get answers from LLMs. Prompt engineering, learning, reasoning etc. are in themselves important and current research topics to investigate and we are planning to do that in near-future work. In particular, we believe that it will be crucial, as we collect enough data, to evaluate learning methods across the full spectrum available from in-context learning, to supervised finetuning, RLHF etc. Again, this is beyond the scope of the current paper.
>
> One unfortunate aspect of tuning individual prompts per LLM is that it is not an exact science and one could always claim that with a slightly different prompt, the output might be even better. To gauge the influence of better prompts we still argue that having a baseline number with a vanilla prompt is important for comparison.
>
> We are grateful to the reviewer for the idea of evaluating a prompt that summarizes the criteria (presentational and epistemological) used to evaluate the answers, possibly with few-shot examples. We believe they would provide a natural extension of the base case. We are trying to run an experiment in that direction. If we can have results in time, we will include them in an updated version of the pdf within the rebuttal period.

---

> > ### Author Response · Authors · 2023-11-22
> > **Response 2 to Reviewer 6F8Y**
> >
> > We sincerely appreciate your time and effort in reviewing our paper, and have now updated it.
> >
> > To address your questions, we ran a preliminary experiment that is adding a description of the criteria to the prompt for GPT-4. The results are included in a new section of the Appendix, A.1.
> >
> > Please let us know if you have any further questions.

---

### Official Review · Reviewer_Y9KD · 2023-11-07

**Soundness:** 2 fair
**Presentation:** 3 good
**Contribution:** 2 fair
**Rating:** 5
**Confidence:** 5

**Summary:**

Given the increasing use of LLMs in contexts where people would previously have used search engines or asked other humans, it is important to understand whether they give accurate information. In particular, artificial controversy and misinformation have slowed and reduced humanity's response to the climate crisis, making it all the more important to investigate the "scientific communication" occurring via use of LLMS. This paper investigates several different LLMs answers to questions about climate change. These are assessed on a 5-pt scale on 8-dimensions (accuracy, uncertainty, completeness, correctness, etc.)  by 3 human raters per answer.  An LLM is also used to generate "assistance" binary answers to the 8-D scale and the usefulness of this assistance is assessed.

**Strengths:**

The paper tackles an important question -- how well do LLMs do at answering questions about climate change? To my knowledge this work is completely original in tackling this question, which is of great significance.
It does a good job of linking to work in assessment of AI systems (e.g. scalable oversight); the related-work section overall is clear and of good quality.
The 8-dimension framework is clearly explained, and abundant resources are cited in the literature informing the framework. There has clearly been a lot of thought put into the framework.
The assessment of the AI assistance is a nice addition.

**Weaknesses:**

This paper is aiming to help us understand whether LLMs give (in)correct answers about climate information, an important and interesting thing to investigate, but almost no consideration is given to the truthfulness or trustworthyness of the answers -- e.g. given prompting about typically-conservative questions, or other random characteristics does the LLM give less accurate answers? The authors state it would be "ideal" for the LLM to tailor responses to the user, but this seems far from obviously ideal in the context of science facts given high levels of LLM sycophancy. Further on the point of LLM correctness, the human evaluators are not climate experts, and there is very little assessment to indicate they would know anything more about the correctness of climate science statements than the supposed "recipient" of the answer. They are assessed by "doing 3 questions fully", but there is no ground truth given, so especially combined with the AI assistance, the whole setup seems like the blind leading the blind.

The main thing I was excited to learn from this paper was whether LLMs are reliable communicators about climate information or not, including specific ways/percent of the time they are unreliable or inaccurate, and I don't understand more having read this paper than I did before.

The rating system is given very little description compared to the 8-dimension scale. What do the numbers 1-5 mean? or what makes a fact "4 accurate" vs 3 or 5 accurate? What guidance was given to raters about these numbers? (their coherence also affects the interpretation of the AI assistance results) It doesn't have to be a philosophical treatise on the nature of truth or something, but maybe focusing on clear facts and evaluating binary accuracy (or binarizing the rating after the fact according to some criteria) could have been an improvement. Overall, I personally only basically care about correctness/accuracy for the purposes of this study; stuff like comprehensibility is a nice add but there are abundant other resources demonstrating the fluency and comprehensibility of LLM responses. and it seems strange/"wrong" to me to elevate "style" to to the same level of consideration as correctness.

The section explaining the 8 dimensions is thorough, but more of a lit review explaining why the dimensions are important for scientific communication (e.g. communicators should tailor to recipients numeracy level) with no connection/adaptation whatsoever to the LLM context overall or this study in particular (e.g. what is the relevance to this work of scientific consensus functioning as a gateway belief? This seems like citation fluff, or at best appropriate for an appendix going into more depth on the dimensions than should be included in the main paper).

Unclear/misleading description of the "Data".  The "answers" are not ground truth answers given by experts, or sourced from wikipedia or something (right?) they're just generated by the LLM. so the keypoints are just ...getting the LLM to choose 25-100% of its own answer as being most important? Why ask the model to copy its own answer verbatim? Or this is a separate post-processing done with a different LLM? This seems unnecessarily complex given the high portion of the "answer" that the keypoints are. Sidenote, the term keypoint was confusing to me here, keypoints are used in computer vision (when dealing with points), usually we would say key phrases or something line that as far as I know (since they're phrases not points), but not a big deal. The post-hoc fetching of "evidence" seems really strange to me. If you make up facts and then search selectively for things that support your made up facts, I would not call that evidence. Maybe injecting the evidence in the prompt or something and measuring how faithful LLMs remain to the injected evidence would be helpful? But measuring evidence this way seems analagous to measuring causality via correlation.

The writing overall is wordy and overselling in a way that compromises clarity and takes up too much space. Overblown language like "treacherous", "meticulously", or "utmost" are out of place in scientific writing, and give the impression of overcompensating/overselling.  The intro in particular is over-broad for what is actually studied (no need for a whole paragraph on public climate literacy). In general, the paper takes a long time to get into details, and these details are not so complex that they can't be quickly mentioned in the abstract and intro. E.g. the pts assessed in the 8-dimension framework. This creates a further obfuscation effect (making the work seem more complex than it is). Show don't tell, and show earlier.

The intro focuses on factuality, but then factuality is not assessed. Instead there is a detailed description of the 8 dimensions, organized into two categories. While each element is exhaustively explained, this could go in an appendix; the words are pretty clear. However despite the verbose explanation, there are no examples, and I'm left not understanding e.g. what is the difference between accuracy and correctness?

AI assistance: The assessment of the AI assistance is supposed to be about providing "scalable" oversight, but there's no analysis of how the assistance scales (or why it would need to in this context), or how much /if it saves the raters any time. E.g. if expert and nonexpert assessment was combined, does assistance enable us to use fewer experts per question? The assessment finds that more issues are raised, but it's unclear if these issues are "real" in the first experiment, and they are baked in in the second experiment (i.e. there are no controls with no issues. So false positives cannot be assessed). Also, a significant issue with using AI assistance for evaluating AI systems is whether that assistance biases the raters in an undesirable way (e.g. distract from subtle issues while raising spurious issues).

Very little analysis and discussion of results, captions unclear (takeaways?), claims made that do not appear supported by the results/experimental design to me (e.g. factuality not assessed; eg. about the utility of AI assistance)

**Questions:**

What is the difference between accuracy and correctness?
What do the 5 levels mean?
Could I see some examples of LLM answers that had the most and least issues?
Could you make some better plots that dig into your results more? I don't have something super specific in mind, but examples of the issues would be a start, maybe breakdown of what types of Qs models are not accurate/correct on, etc.

I initially rated this paper as borderline, because I think the question you're tackling is valuable, but especially on getting to this section, I realized it would take a substantial overhaul of all the experiments section (intro stuff and setting up rating framework should be about 1/2length IMO, and the experiments and discussion of results at least doubled), and probably further experiments with experts, not just random humans, a refocus on factuality, and substantial writing changes, in order to make it something I believe is above the bar for publication. I strongly encourage the authors to pursue a resubmission with improvements, and I'm willing to revise my score if at least substantially more analysis is done in the review period.

---

> ### Author Response · Authors · 2023-11-15
> **Response to Reviewer Y9KD (Part 1/2)**
>
> We thank the reviewer for their comments.
>
> First we want to clarify a few misunderstandings, we will also do our best to make them clearer in the paper itself:
> * **Raters:** We mention in the main text that raters have relevant educational background, and give very detailed information in the appendix (Appendix M): they have "at least an undergraduate degree in a climate-related field of study. This includes environmental disciplines (e.g. environmental science, earth science, atmospheric physics, ecology, environmental policy, climate economics), and also other disciplines (including the behavioral and social sciences)." Moreover, after completing a tutorial, the raters have to pass a test where they complete three full questions to determine whether they have a good enough grasp of the task.
> * **Key points:** The key points are used in two different ways. 1) In the attribution evaluation (Section 4) to break down the answer into key statements,  making it easier for the raters to identify support/refusal by the given evidence. 2) Moreover, the evidence is used as additional information to ground model assistance on the epistemological dimensions.
> * **Correctness vs Accuracy:** (Linguistic) correctness is a presentational dimension that captures spelling mistakes and grammatical errors. Accuracy on the other hand is an epistemological dimension that captures whether the information provided in the answer is accurate. We will try to emphasize the difference in the paper even more.
> * **Rating scale:** In the paper we say: “In our interface the raters agree with a statement (see Table 18) on a 5-point Likert scale between disagree completely to neither to agree completely which we map to 1 . . . 5. See Figure 15 for a screenshot.” So {1, 2} means the rater disagrees with the statement (and is then asked to add further details by selecting issues), while {3} means neither agree nor disagree, and {4, 5} means they agree with the statement. This is a common and established setup in human studies.
> * **Scalable oversight:** This is a technical term in the area of AI alignment and should be interpreted in that context. We cite the relevant literature in the paper.
>
> We want to emphasize one point especially, since it seems to lead to a fundamental misunderstanding of the epistemological part of our evaluation. In the review it is stated that “factuality is not assessed”. We are puzzled by this and in the absence of a more precise definition of ‘factuality' it may not be possible for us to provide clarifications. Would you be willing to provide us a definition of  ‘factuality’ and clarify why you think it is not assessed by our evaluation, in particular the epistemological accuracy dimension?
>
> We do analyze types of questions with regard to dimensions. We report in the main body of the paper (section 3.3, page 6, other findings) and Appendix I that generally we did not find a significant impact of question type on the ratings. However, we do find that the synthetic Wikipedia questions tend to receive lower epistemic scores. Furthermore, we have found that questions on “Policies-Mitigation-Adaptation” tend to score lower in “tone” dimension.
>
> Lastly, you mention that you would like to see more analysis and visualizations, and would like us to reduce text explaining the science communication background. We believe the interdisciplinary aspect of the work, grounded in Science Communication research, is a core contribution of the work and we do not intend to reduce this part. We also kindly but firmly push back against the insinuation that this is “citation fluff”. It is the theoretical foundation of this work, and it was prioritized deliberately. We similarly push back on the remarks about ‘overblown language’, we find those comments not conducive to a constructive, respectful conversation.
>
> We also emphasize that we provide extensive analysis and visualizations. The paper is already very long with 49 pages. The examples that you ask for are actually provided (“examples of the issues would be a start”: Table 18 has all statements and issues, Appendix N has many examples including issues; “maybe breakdown of what types of Qs models are not accurate/correct”: Appendix N contains examples, specifically Table 23 and Table 28 contain examples of an inaccurate answers).

---

> > ### Author Response · Authors · 2023-11-15
> > **Response to Reviewer Y9KD (Part 2/2)**
> >
> > We do include specific good/bad examples for dimensions in Appendix N. There is no optimal answer to a question, and there are many different ways to answer a question while not exhibiting any science communication related issues. Here is an example of an overall good and an overall bad answer as measured by our evaluation:
> >
> > Bad answer:
> > ```
> > Question: Is Pluto warming?
> > Answer: Yes, Pluto is warming. This is due to the fact that Pluto is not a planet. Pluto is a dwarf planet. It is a small icy world that orbits the sun. Pluto is also the farthest planet from the sun. This means that it takes Pluto a very long time to get close to the sun. This is why Pluto is so cold. Pluto is about 4.5 billion miles from the sun. That is a long way away. Pluto is so far from the sun that it takes Pluto 248 years to orbit the sun. Pluto is also a very small planet.
> >
> > There are issues for all dimensions:
> >   [presentation] [2, 2, 3] Issues: ['repetitive', 'repetitive']
> >   [understandability] [1, 3, 3] Issues: ['other', 'Unclear ']
> >   [correctness] [1, 2, 3]  Issues: ['other', 'Contradictory ', 'other', 'contradictions ']
> >   [tone] [2, 2, 3]  Issues: ['biased', 'biased']
> >   [accuracy] [1, 1, 3]  Issues: ['incorrect', 'incorrect']
> >   [specificity] [1, 2, 3]  Issues: ['other', 'Incorrect ', 'irrelevant_info', 'vague']
> >   [completeness] [1, 1, 3]  Issues: ['does_not_address_main_parts', 'not_enough_detail', 'not_enough_detail', 'ignores_science']
> >   [uncertainty] [1, 1, 2]  Issues: ['uncertainty_missing', 'consensus_missing', 'uncertainty_missing', 'uncertainty_missing', 'consensus_missing']
> > ```
> >
> > Good answer:
> > ```
> > Question: How is climate different from weather
> > Answer: Climate refers to the long-term patterns and trends of temperature, precipitation, and other weather conditions in a specific region. It is determined by factors such as latitude, elevation, and ocean currents. Weather, on the other hand, refers to the short-term and day-to-day changes in temperature, precipitation, and other atmospheric conditions. While weather can change quickly and vary greatly within a single day, climate remains relatively consistent over longer periods of time.
> >
> > No issues at all:
> >   [presentation] [4, 5, 3]  Issues: []
> >   [understandability] [4, 5, 4]  Issues: []
> >   [correctness] [4, 5, 4]  Issues: []
> >   [tone] [4, 5, 4]  Issues: []
> >   [accuracy] [5, 4, 3]  Issues: []
> >   [specificity] [4, 5, 4]  Issues: []
> >   [completeness] [4, 4, 4]  Issues: []
> >   [uncertainty] [4, 4, 4]  Issues: []
> > ```
> >
> > In case there is some other example missing that would help the presentation please let us know.

---

> > > ### Author Response · Authors · 2023-11-22
> > > **Response 2 to Reviewer Y9KD**
> > >
> > > We sincerely appreciate your time and effort in reviewing our paper.
> > >
> > > In our updated version of the paper, we have simplified the introduction and the discussion of the theoretical concepts behind the dimensions, specifically addressing a major point you made in your review.
> > >
> > > Please let us know if you have any further questions.

---

### Official Review · Reviewer_ihxg · 2023-11-10

**Soundness:** 2 fair
**Presentation:** 4 excellent
**Contribution:** 3 good
**Rating:** 5
**Confidence:** 5

**Summary:**

This paper provides a framework to evaluate the responses of LLMs to climate change questions (based on both presentational and epistemological adequacy). The paper further analyzes the performance of seven different LLMs on these criteria, using a dataset of 300 questions (obtained from Skeptical Science, Google Trends, and GPT-4 generation of questions on English Wikipedia articles), and according to ratings provided by a group of 32 raters (three raters per question).

**Strengths:**

Topic/relevance: This submission addresses a very important topic. While there is genuine debate and disagreement as to whether LLMs should be used to serve climate change related information, the reality is that they are being used to do so in practice. Evaluation metrics are therefore key to assessing the quality and potential pitfalls of this information-serving, in order to suggest guardrails, improvements, or avoidance of use as applicable.

Criteria introduced: The evaluative dimensions presented in the paper span important axes of both presentation and epistemological adequacy, and I believe the authors hit many of the needed points in this regard.

Methodology for evaluation: The scope and execution of the evaluation is impressive, with meticulous documentation of criteria for gathering and filtering questions, solid criteria for training raters and evaluating (dis)agreements among them, thorough analysis of results, reporting of error bars, and inclusion of raw data and examples in the appendix. I am very appreciative of the transparency and thorough documentation of the process and results. The diversity and global spread of raters is also impressive, and I appreciate the authors' practice of compensating raters properly.

Evaluation: The concrete evaluation of several models provides useful insights on the overall performance (objective and comparative) of current models. If properly contextualized, this evaluation can provide useful insight to practitioners evaluating whether or how to use LLMs for the purposes of serving climate information.

**Weaknesses:**

Before sharing my impression of weaknesses, I would like to emphasize that I really do commend the authors on this line of work, as I think it is extremely important (as noted above). It is precisely because of the importance/applicability of and attention to this kind of work that I present the feedback below, as I believe it is exceedingly important that users of this work (including individuals from ML, climate, and/or general backgrounds) are able to appropriately understand and act on the results based on a solid understanding of the limitations/caveats of the analysis. I hope the authors are able to provide appropriate discussion of these limtations/caveats based on the feedback below.


Presentation of scope: The submission is presented as addressing "LLM responses to climate change topics." However, the scope is in reality much narrower than that - it is seemingly more along the lines of "LLM responses to general questions on climate change for the climate-curious general public" rather than "LLM responses to specific questions that may be used by a climate change decision-maker or practitioner to take action." This is evidenced, e.g., by the choice of question sources (notably, the focus on Skeptical Science and Google Trends) and the Specificity Filter used to remove questions that ask about "a very specific and narrow topic" (Appendix C). This distinction in scope, while subtle, is actually exceedingly important, as the stakes (and therefore, the criteria / weighting of criteria used) are very different in general-curiosity vs. action-oriented settings. To the extent to which this study will be used as a snapshot of the current state of quality of LLM responses for climate, it is critical to clarify the kinds of use cases this evaluation is actually applicable to (given the range of use cases today from, e.g., general-public-Q&A like ChatClimate and ClimateQ&A, to action-oriented models attempting to e.g. give information on climate adaptation practices to farmers who would actually implement those practices).

Lack of discussion of rationale for design of LLM answer structure: The authors present the protocol for getting answers, keypoints, evidence, and AI assistance from the LLM - but the rationale for using this protocol is not discussed. Is the goal to get as good a model as possible, a model that is representative of what's out there, to test out raw output vs. summarization vs. self-correction capabilities, etc.? Discussion of these design choices is critical to properly contextualize the evaluation of the outputs.

Implicit weighting of criteria: Throughout the discussion of evaluative dimensions and their presentation in the numerical study, there is an implicit assumption that all criteria are weighted equally. However, different criteria can and often do have different weights - for instance, with respect to epistemological adequacy, if accuracy is not met, high performance on specificity/completeness/uncertainty is arguably irrelevant. This is not addressed in the submission at all, and scores are often averaged for the purposes of visualization and analysis. Similarly, error bars are presented to give a sense of uncertainty, but extremal measures (worst-case and best-case outcomes) are not reported - however, in settings where the answers may be "safety-critical" (e.g., the example of telling a farmer what practices to implement, where an incorrect answer has the potential to damage a farmer's entire crop and livelihood), worst-case outcomes in particular are key to have a handle on.

Choice of raters: The cohort of 32 raters chosen for the study is described as consisting of individuals with "at least an undergraduate degree in a climate-related field of study. This includes environmental disciplines (e.g. environmental science, earth science, atmospheric physics, ecology, environmental policy, climate economics), and also other disciplines (including the behavioral and social sciences)." I have some concerns about this choice of raters. First, there is some mismatch between the kinds of disciplines listed and the disciplines of the questions (see Figure 5) - topics like cities-settlements-infra, energy, and health-nutrition are covered in relatively different disciplines than those listed. Second, answers can be inaccurate in subtle but pernicious ways, which requires targeted expertise to catch - it is not clear that raters were necessarily matched to questions in their area of expertise, nor that they were given time to consult external literature (e.g. by performing their own Google Search or literature search) to fully evaluate the accuracy or broader epistemological quality of answers. For context, I personally have higher than an undergraduate degree in a climate-related field of study (i.e., meet the criteria for inclusion as a rater), but would not be able to confidently answer questions about the accuracy of information on e.g. pest spread (Table 6) without consulting external sources, given that this is outside the climate sector I am most familiar with and given that such answers can be wrong in subtle ways that I may not immediately catch (e.g. warmer conditions have different effects on different pests, which means the effects can vary geographically and blanket statements may be incorrect). This concern is substantiated by observations such as low inter-rater agreement on specific answers (Appendix H), and while the authors claim that system-level conclusions can still be drawn on the basis of the results, I do not think this is categorically true and it warrants more nuanced discussion. While I acknowledge it would be unfair to ask the authors to redo the entire study, these issues of rater expertise should be foregrounded in the main body of the paper, and be taken into account via appropriate caveats on results (including the results on the effectiveness of AI assistance - see next point).

Analysis of AI assistance: Related to the previous point, the effects the authors describe with respect to the helpfulness of AI assistance may actually be more of an indication of the lack of matched expertise of the raters, rather than being a clear indication of whether AI assistance is actually helpful in reality or not (especially given lack of access to ground truth in the context of this study). In addition, results indicating that AI assistance helped raters find more mistakes may also be a result indicating phenomena more along the lines of "anchoring and adjustment" rather than the true presence of more mistakes in the answers (especially given that, as shown in Figure 3a, the biggest boost actually came in specificity, completeness, and uncertainty - which are more subjective - rather than accuracy).

Next steps and takeaways: While the goal of the paper is presumably to motivate potential next steps and/or cautions regarding the development and use of LLMs for answering questions about climate change, it is notable that discussion of critical recommendations, next steps, or hopes for usage of the presented framework are missing. Explicit discussion is necessary to concretize the "point" of this paper, and provide further context for the analysis/design choices therein.

Additional comments:
* Section 2.2, Specificity: The authors state, "if a question is posed about climate change in India, the reply should provide data and insights relevant to the Indian context." This is a nuanced and potentially controversial statement (à la the critiques on "filter bubbles"), and this nuance/debate should be acknowledged. (It is also worth noting that higher specificity may lead to higher trust *even in situations when the answer is inaccurate*. As a result, "should" is a loaded word here - if the answer is inaccurate, maybe it is better that the model provide an answer that is otherwise presentationally and epistemologically low-quality in a way that makes the user *distrust* the answer.)
* The prompt used for the LLMs specifies the LLM is an "expert on climate change communication" - notably, not, e.g., an "expert on climate change who is good at general public communication." In other words, the current choice of prompt implies a particular weighting between presentational vs. epistemological criteria, and a different prompt would imply a different rating. While I again acknowledge it would be unfair to ask the authors to redo the entire study by rewording the prompt, the potential implications of the current prompt should at least be highlighted explicitly.

Minor:
* Figure 1(b): It is important to flag explicitly (e.g. via a cut y-axis) that y-axis does not start at zero.
* Figure 3: Would it be possible to add error bars to this figure?
* Appendix, Table 6 (page 24): "Pragraph" --> "Paragraph"

**Questions:**

* Is it possible to update the body of the submission to further clarify the scope of the evaluation and add caveats accordingly, in addition to potentially sharing the exact questions that were evaluated?
* What is the rationale for the design of the LLM answer structure?
* Is it possible to present the evaluative dimensions and numerical results in a way that discusses not only average performance across all criteria, but is also informed by potential weightings of criteria and considers extremal (not just average) performance?
* How might the analysis of results change given the observations above on the expertise of raters, and what caveats or re-interpretation of numerical results are needed accordingly?
* What are the next steps and major takeaways from this work?

---

> ### Author Response · Authors · 2023-11-15
> **Response to Reviewer ihxg (Part 1/2)**
>
> We sincerely thank the reviewer for the in-depth review and their specific and thoughtful comments and suggestions.
> First we would like to state clearly that by and large we agree with the points raised. Several of these points (and more) have been, and will continue to be, the object of internal discussions among the authors themselves. They shape how we approach this work which possibly is the first to explore this topic so directly. As a consequence, there is little to build upon. Accordingly, we made design choices and deliberate compromises that will have to be revised in the future, or will inspire other research teams to contribute to this research question using different approaches. For the moment, our design choices served the purpose of providing a first holistic picture on the topic. We are happy to engage in this conversation, to clarify and improve where possible, and otherwise inform our future work.
>
> To the specific points:
>
> **1- Scope**
>
> Coming up with a general and representative set of questions is in itself not a trivial task. There are no standardized tests for evaluating climate change-related knowledge. The choice of the current set is based on the observation that these are questions users care about and frequently ask about (Google Trends), have been debated for decades and used to influence the public (SkepticalScience), and Wikipedia articles have been written about to answer them (Wikipedia).
> These are by no means the most important, or most useful, or hardest questions on the topic. But they provide a natural starting point: it seems reasonable to understand how LLMs respond to them before going deeper into more specific topics.
> Once we have convinced ourselves, and hopefully the community, that the foundation is solid, we will move to harder, more specific and actionable questions. We want to release the dataset and are working towards that end.
> Note: we excluded exceedingly specific/narrow questions from Wikipedia because we felt confident that no expert would be able to answer them without the knowledge and presence of the specific article they were generated from. In other words, that task would have been closer to a machine reading one.
>
> More broadly, we envision our evaluation framework to evolve to cover all relevant use cases related to climate information.
>
> **2- Design rationale**
>
> The design of the LLM task was chosen so that it is a good match for the main use case we are addressing: The general public engaging with LLMs to satisfy climate change related information needs. We will highlight the assumed audience more explicitly. We’re trying to keep the setup as simple as possible and realistic at the same time. One reason for adding a soft limit to the answer length is to avoid (for the time being) adding confounding factors related to answer structure, length, and complexity. This helps to make answers comparable between models, which otherwise can produce much longer replies which could bias raters.
> We also didn’t want to make use of complex or model-specific prompts, e.g., reasoning-based, because they may be interpreted and executed in different ways by different models. The study of the effect of prompt engineering and reasoning is a very important topic, but would introduce too much complexity. We do however plan to investigate this extensively in future work. We believe our somewhat straightforward prompt is agnostic to the specific model and doesn't add relevant bias.
> We found that keypoints and url generation provide a nimble first solution to adding evidence that can be used to 1) inform AI assistance, and 2) provide support for running the AIS evaluation (which needs references). Note that these steps are executed after the answer is generated, and always by the same model (GPT-4), so they don’t affect the answer quality, which is meant to represent the LLM’s ‘vanilla’ answer mode.
> Overall, we iterated over multiple base versions to get to a simple and robust initial ‘baseline’ mode. We expect that models will perform better in more sophisticated modes, but it would introduce more complexity than appropriate at this stage. Overall, the high-level design choice is that model and evaluation sophistication must progress in tandem.
>
> **3- Weighting of Criteria**
>
> We agree that the combination of scores is a critical problem for making decisions - which we don't. Averages in Fig. 1 are mainly given to illustrate the point that LLMs overall do much better in the presentational dimensions. For most of our analysis we report the individual dimensions. We agree that deciding which LLM is 'best' for the task might require some heuristic aggregation of the different dimensions. We deliberately do not propose an ad-hoc heuristic.

---

> > ### Author Response · Authors · 2023-11-15
> > **Response to Reviewer ihxg (Part 2/2)**
> >
> > **4- Choice of raters.**
> >
> > In our current approach we struck a compromise between availability / cost of ratings and specific domain expertise. If we require too specific domain expertise, recruitment problems will lead to issues regarding scalability and replicability. In future work, we plan to explore the furthest end of this by working with domain experts and matching them to questions based on topic, also adding a reconciliation step where the domain experts can resolve divergences. We hope it is clear that this goes far beyond what we are currently doing here.
> >
> > Notice that we don’t forbid or discourage consulting external sources, we only specify that this should be done quickly to clarify specific points rather than to perform an in depth research of the topic, because this may not only make the task less scalable, but also introduce other confounding factors. However, this is an aspect of the research which we are actively reviewing and will evolve as we identify better solutions. Again, we don’t feel this choice invalidates the findings of the current work.
> > At the same time, we fully agree with the reviewer that appropriate caveats and a more nuanced presentation of take homes from the results should be presented more clearly. We will address this in the revised version of the paper.
> >
> > **5- AI Assistance.**
> >
> > We agree that our results about AI Assistance only apply to the specific setting we're using it in, and can’t be generalized to a setting where domain experts are used as raters on matched questions. We do however believe that the setting we’re studying is of interest, because a group of raters with relevant background is much more accessible than a group of domain experts.
> >
> > We do in fact want to explore the question whether domain experts would also benefit from AI Assistance, but it's out of scope for this paper and something we have to leave for future work.
> >
> >
> > **6- Next steps and takeaways.**
> >
> > We will add a paragraph on this.
> >
> > Section 2.2
> > This is a good point. It ties back to the way such evaluation should be used and how the individual dimensions are combined and weighted. Consider the following example: “is planting trees good for climate change?” Maybe a baseline weight should be ‘accuracy trumps everything else’. But it is not so easy. For instance, it is probably accurate to say that planting trees can be good for climate change, because they are natural carbon capture and storage. However, research also suggests that if done badly (e.g., due to differential albedo effects in equatorial vs boreal regions) it can do more harm than good. Thus, completeness must also be taken into account. But, then, uncertainty may need to be factored in, leading to different conclusions… and so on and so forth. There are no shortcuts and easy answers, an expert would take all these dimensions into account to make a comprehensive judgment.
> > These are key observations and limitations of our research which we are not in a position to resolve, the reviewer is absolutely right, but we feel they go beyond the scope of the current work.
> > These are observations that must be articulated clearly at the beginning and at the end, whenever takeaways are presented.
> >
> > **7- Prompt.**
> >
> > We evaluated many variants of prompts and targeted the simplest most reasonable configuration. However, the effect of prompting is complex and deserves a thorough investigation that goes beyond this paper.

---

> > > ### Comment · Reviewer_ihxg · 2023-11-21
> > > **Response to authors**
> > >
> > > Thank you to the authors for their thoughtful response. To respond to the specific points:
> > > * **Scope:** I agree that it is nontrivial (and perhaps impossible) to come up with a general and representative set of questions. My suggestion was not that the set of questions be changed, but rather that the paper needs to much more clearly signpost and clarify the ways in which the analysis and its conclusions are affected by the choice of questions (notably, this choice has significant implicit and explicit effects on model performance and on the choice/weighting of performance metrics). Otherwise, there is nontrivial danger that the results of this paper will be read and used out of context.
> > > * **Design rationale:** Thank you for elaborating further– the points on audience and prompt length make sense, and I look forward to this being spelled out more explicitly in a revision of the paper. I still do not fully understand the rationale for the workflow of keypoints, URL generation, etc. – while they may not affect the base answer, they are likely to have substantive effects on how raters evaluate the quality of the responses (whether positive or negative). Given the rater-based nature of the study, these design choices do merit much more explanation.
> > > * **Weighting of criteria:** Thank you for the clarification, and I appreciate that many figures throughout the manuscript show disaggregated results. That said, the authors should take care that result summaries in the text do not also impose an implicit weighting on criteria – e.g. statements like the following place an even weighting across all disaggregated criteria, across epistemological and presentational attributes: “We found that the majority of three raters detected 77% of issues when shown assistance, while the majority of three raters only detected 60% of the issues when not shown assistance.”
> > > * **AI Assistance:** I do think the points about AI Assistance need to be treated with much more nuance. Notably, it is not currently clear whether the goal is to discuss whether AI Assistance can be helpful in the context of doing ratings-based studies like this (be it with the current group of raters or domain experts), or whether the idea is to suggest AI Assistance as an add-on to how information is displayed the public. (In other words, is this a takeaway for other researchers studying the efficacy of LLMs, or is the takeaway for deployers and users of LLMs?) The use of AI Assistance has very different implications in these contexts.
> > > * **Other points:** I am satisfied with the authors’ responses to the remainder of my points.
> > >
> > > Overall, I appreciate the authors’ reflectiveness. That said, since the authors have not uploaded a revised version of the paper, it is hard to tell how these changes will be implemented in practice. As such, I have raised my rating from 3 to 5, but am not yet able to recommend acceptance.

---

> > > > ### Author Response · Authors · 2023-11-22
> > > > **Response 2 to Reviewer ihxg**
> > > >
> > > > We again sincerely thank the reviewer for their insightful comments; we truly believe they lead to an improved paper that is much clearer about what it does and what its limitations are (see updated version).
> > > >
> > > > Regarding the specific points:
> > > >
> > > > * **General:** We added a paragraph at the end of the introduction and in general used a more cautious language about our analysis and conclusions throughout the paper.
> > > > * **Scope:** We have added a section (3.1) that specifically addresses the scope of the questions we assess and provides rationale for it.
> > > > * **Design rationale:** We’re now spelling this out more explicitly throughout the paper; note particularly the changes to the paragraphs discussing the answer, evidence and keypoints in Section 3.2. We’re also adding to the explanation for keypoint and URL generation – these are not suggestions for an optimal way to retrieve information, but rather a simple baseline protocol (that in our early experiments was comparable to using an off-the-shelve retrieval system, but much simpler).
> > > > * **Weighting of criteria:** Thank you, we did our best to add further context when making statements such as the one you cite, to make sure we do not imply an equal weighting would be a good choice.
> > > > * **AI Assistance:** We added nuance to the places where we discuss the AI Assistance, and specifically make clear that the goal of AI Assistance in this study is to assist raters (and not provide additional commentary when displaying the answer in public).
> > > >
> > > > We really appreciate the time and effort that went into your review.

---

> > > > > ### Author Response · Authors · 2023-11-23
> > > > > **Response 3 to Reviewer ihxg**
> > > > >
> > > > > We just updated the paper again, making changes to how we discuss "Resolution and Range" in Section 3.4.
> > > > >
> > > > > This is is relevant to observations made in your review and we draw your attention to it as we have now removed any remaining suggestions about conclusion drawn from aggregated scores. In the updated version we provide more nuance and accurate remarks, particularly with respect to what we mean by resolution and range.

---

### Meta-Review · Area_Chair_WPeR · 2023-12-09

**Metareview:**

This paper attempts to measure the reliability of LLMs in providing information related to climate change. This is a highly important topic which the reviewers agree is tackled thoughtfully here. However, the reviewers largely agree that the methodology has some serious flaws that prevent it from justifying the authors' conclusions, and also that the presentation of the paper is liable to severe misinterpretation with potentially serious consequences. I therefore encourage the authors to consider this feedback and redesign their experiments with it in mind, but must recommend rejection of the present paper.

As a note to the authors, reviewer ihxg had the following feedback to give following the revision (due to the OpenReview platform restrictions, this final response was unfortunately not visible to the authors). There is a response from Y9KD as well (which I hope the platform will make available to the authors later but which this box is unfortunately too small to contain).

- - -

Thank you to the authors for their response and revisions to the manuscript. The clarifications added to the manuscript definitely present an improvement over the previous version. However, I still have several concerns about the present version of the manuscript, which would need to be addressed in order for me to raise my score further.

Scope: I appreciate the new clarification on scope in Section 3.1. However, I think this section lacks necessary depth; it reads more like a statement of "we are not focusing on / cannot focus on all questions, so we focus on popular forums" rather than a nuanced discussion of the ways in which this choice of scope may affect the analysis. Likewise, discussion of the implications of the choice of scope is not currently integrated into the analysis. As mentioned previously, without this nuanced discussion, there is a real danger that the conclusions of this paper will be taken out of context (e.g., in evaluating and/or justifying the use of LLMs for serving climate information in safety-critical contexts).

Design rationale: I still do not think the design rationale is spelled out explicitly enough. For instance, in Section 3.2, the following statement appears: "We found that using keypoints, in combination with URL generation and evidence selection, is a simple and effective solution, compared to a paired retrieval components, for our current setup." What is using keypoints/URL generation/evidence selection a simple and effective solution for - providing context for raters, providing context for the general public, shedding light on LLMs' evidentiary support, or something else? This is just one example of where I do not think the design rationale is clearly enough spelled out. The manuscript needs much more context on which design decisions are made for the benefit of the authors vs. raters vs. users, and what the implications of those design decisions might be.

AI Assistance: I appreciate the clarification added in the manuscript that the goal of AI Assistance is to help human raters. I think it would be important to add some further clarification and caveats as to why AI Assistance was helpful (e.g. talking about the expertise of the raters and why/whether the results may change with more expert raters).

Next steps and takeaways: I still do not think my original comments are addressed in the revised manuscript.

I appreciate the updates in the intro regarding weighting of criteria (that said, I think this could also be spelled out in more detail in Section 2 where the criteria are introduced) and the emphasis on evaluating criteria in a disaggregated manner.

**Justification For Why Not Higher Score:**

The reviewers largely agreed on rejection.

**Justification For Why Not Lower Score:**

n/a

---

### Decision · Program_Chairs · 2024-01-16

Reject